# UniFast-HGR: Scalable and Efficient Maximal Correlation for Multimodal Models

## Abstract

This paper introduces UniFast-HGR, an efficient and scalable framework for estimating Hirschfeld-Gebelein-Rényi maximal correlation in multimodal learning. The method addresses computational bottlenecks in traditional HGR and Soft-HGR approaches, which suffer from $O(K^3)$ complexity due to covariance matrix inversion and limited scalability to deep architectures. UniFast-HGR incorporates three key innovations: replacing covariance with cosine similarity to avoid matrix inversion, removing diagonal elements to mitigate self-correlation bias, and applying $\ell_2$ normalization as a variance constraint for improved stability. These improvements reduce computational complexity to $O(m^2 K)$ while maintaining bounded correlation scores. The OptFast-HGR variant further accelerates computation by simplifying normalization steps, achieving dot-product-level efficiency with minimal accuracy loss. Experimental evaluations across benchmark datasets validate the framework's ability to balance computational efficiency with accuracy, establishing it as an effective solution for addressing contemporary deep learning challenges.

## 1 Introduction

Learning effective representations from data is a central challenge in machine learning (Bengio et al., 2013). This challenge is significantly amplified when dealing with multimodal data, which integrates information from diverse sources such as images, text, and audio (Summaira et al., 2021). While human cognition inherently fuses these disparate signals for robust understanding, artificial systems often struggle to synthesize heterogeneous modalities. A primary difficulty stems from the distinct statistical properties, noise characteristics, and dynamic ranges inherent to each modality, which can obscure the underlying cross-modal dependencies crucial for learning joint representations (Baltrusaitis et al., 2018; Guo et al., 2019; Gandhi et al., 2023). Contemporary multimodal learning frameworks employ various alignment mechanisms, including contrastive objectives based on pairwise similarity, cross-modal attention within transformer architectures, and mutual-information-inspired criteria. Although these paradigms deliver strong performance, particularly in large-scale settings, they predominantly optimize for sample-wise correspondences and often lack explicit regularization of richer, higher-order dependency structures across modalities. This limitation can manifest as sensitivity to modality-specific scale mismatches and noise, potentially leading to suboptimal characterization of cross-modal relationships in challenging scenarios involving missing data, label scarcity, or heterogeneous noise distributions.

The Hirschfeld-Gebelein-Rényi (HGR) maximal correlation offers a principled framework for quantifying nonlinear statistical dependence between random variables, generalizing classical linear correlation measures (Hirschfeld, 1935; Gebelein, 1941; Rényi, 1959). Unlike traditional methods such as Canonical Correlation Analysis (CCA), which identifies linear relationships (Hotelling, 1936), HGR maximal correlation provides a mathematically grounded objective for extracting maximally informative features capable of capturing complex nonlinear dependencies. This theoretical strength has motivated its application in multimodal representation learning (Huang et al., 2017). However, the practical integration of HGR maximal correlation into modern deep learning pipelines faces significant computational and numerical hurdles. Let $N$, $m$, $K$, and $M$ denote the number of samples, the per-device batch size, the feature dimension, and the modality count, respectively. Classical HGR formulations impose strict whitening constraints, requiring centered features with identity covariance. This necessitates manipulating and inverting $K \times K$ covariance matrices. When imple-

mented within a deep network's training loop, these operations introduce a per-update computational cost that scales as $\mathcal{O}(mK^2)$ for covariance construction, coupled with a $\mathcal{O}(K^3)$ term for the requisite matrix decomposition. Furthermore, these steps are prone to numerical instability when $K$ is large or when mini-batch covariance estimates are ill-conditioned, posing a severe bottleneck for high-dimensional embeddings common in modern architectures. Efforts to create more tractable approximations have emerged. The Soft-HGR framework relaxes strict whitening constraints via empirical approximations, enabling practical use in applications like audio-visual emotion recognition (Ma et al., 2021) and multimodal correlation analysis (Shi & Huang, 2023). Nonetheless, Soft-HGR and related variants (Wang et al., 2019) still rely on covariance-scale computations with $\mathcal{O}(mK^2)$ complexity and can struggle with efficiency and stability when applied to contemporary models with high-dimensional embeddings trained on large-scale datasets.

The rise of multimodal foundation models and contrastive learning paradigms has further intensified the need for scalable, stable, and expressive dependence measures. Modern systems demand correlation estimators that remain efficient with increasing feature dimensions $K$, robust to mini-batch stochasticity and modality imbalance, and capable of capturing meaningful nonlinear relationships. While existing measures such as centered kernel alignment (CKA) (Kornblith et al., 2019), distance correlation (dCor) (Zhen et al., 2022), and its intrinsic variant ($I_d$Cor) (Basile et al., 2025) offer valuable alignment signals, they often incur substantial computational overhead or face limitations in balancing scalability with expressive power under practical deep training constraints.

To address these challenges, this paper introduces UniFast HGR, an efficient and scalable framework for HGR maximal correlation estimation tailored for multimodal deep learning. The proposed method circumvents key computational bottlenecks through three core design innovations: (1) replacing explicit covariance computation and decomposition with operations based on cosine similarity, thereby eliminating matrix inversion; (2) removing diagonal elements from the correlation matrix to mitigate trivial self-correlation bias prevalent in high-dimensional spaces; and (3) enforcing stable variance constraints via $\ell_2$ normalization. Collectively, these modifications transform the estimator's dominant computational cost from covariance-based operations to Gram/similarity-type computations with $\mathcal{O}(m^2K)$ complexity. This is particularly advantageous in the common regime where the feature dimension $K$ significantly exceeds the batch size $m$. UniFast HGR functions as a modular, plug-in correlation objective that operates on intermediate or final network embeddings. It integrates seamlessly as an auxiliary regularizer alongside primary task losses or contrastive learning objectives. Its design promotes robustness in scenarios with modality imbalance, missing data, or limited supervision. An optimized variant, OptFast HGR, further reduces computational overhead through streamlined normalization, achieving efficiency comparable to standard dot-product operations while maintaining competitive accuracy with bounded approximation error. The main contributions of this work are summarized as follows:

**Efficient and Scalable Correlation Estimation**: UniFast HGR introduces a reformulation of HGR maximal correlation that replaces covariance-based decomposition with cosine-similarity operations under explicit variance constraints. This reduces the dominant computational complexity from $\mathcal{O}(K^3)$ to $\mathcal{O}(m^2K)$, enabling practical correlation estimation for high-dimensional embeddings in large-scale multimodal learning.

**Enhanced Robustness and Stability**: Through diagonal removal and variance normalization, UniFast HGR mitigates self-correlation bias and improves numerical stability in high-dimensional feature spaces. The framework demonstrates robust performance under challenging conditions including missing modalities, label insufficiency, and noisy representations.

**Broad Applicability and Integration**: UniFast HGR serves as a drop-in correlation module compatible with various multimodal learning paradigms and modern neural architectures, including convolutional networks, vision transformers, and foundation models such as CLIP, DINOv2, and ViCLIP. Extensive evaluations across diverse tasks confirm consistent performance gains while maintaining computational efficiency.

**Optimized Variant for Practical Deployment**: The OptFast HGR variant further reduces computational overhead through streamlined normalization, achieving efficiency comparable to dot-product operations while maintaining competitive accuracy. This enables practical deployment in resource-constrained environments and large-scale training scenarios.

These contributions advance the application of HGR maximal correlation to contemporary multimodal learning, providing a scalable, stable, and principled method for enhancing cross-modal alignment in modern deep learning systems.

## 2 PROPOSED METHOD

The UniFast HGR framework significantly improves upon both Soft-HGR and the original HGR maximal correlation approaches by addressing computational challenges, scalability limitations, and practical constraints in large-scale neural network applications. This framework enhances both discriminative and correlation capabilities, facilitating the extraction of highly informative features across diverse data modalities. The following sections outline its key components and innovations.

### 2.1 PRELIMINARIES

**HGR Correlation Analysis and Limitations**: HGR maximal correlation extends Pearson correlation by providing a more comprehensive measure of dependency between random variables. For random variables $X$ and $Y$ with joint distribution over domains $\mathcal{X}$ and $\mathcal{Y}$, given $m \times K$ feature matrices $f = [f_1, f_2, \cdots, f_m]^T$ and $g = [g_1, g_2, \cdots, g_m]^T$, where $f_i$ and $g_i$ are both $1 \times K$ dimensional vectors, $m$ is the batch size, and $K$ is the feature dimension, the HGR maximal correlation is defined as:

$$\rho^K(X, Y) = \sup_{\substack{f:\mathcal{X}\to\mathbb{R}^K, \ \mathbb{E}[f]=0, \ \text{cov}(f)=I \\ g:\mathcal{Y}\to\mathbb{R}^K, \ \mathbb{E}[g]=0, \ \text{cov}(g)=I}} \mathbb{E}[f^T(X)g(Y)] \tag{1}$$

where $\mathbb{E}[f]$ and $\mathbb{E}[g]$ are the expected values, and $\text{cov}(f)$ and $\text{cov}(g)$ denote the covariance matrices.

The HGR correlation ranges from 0 to 1, indicating complete independence or deterministic relationships, respectively. However, the computational complexity arises primarily from whitening constraints requiring matrix inversion and decomposition, resulting in $O(K^3)$ time complexity. These challenges are compounded by scalability issues, as covariance matrices can become ill-conditioned in high-dimensional spaces.

Soft-HGR addresses some computational challenges through low-rank approximations, enabling integration with neural networks without strict whitening constraints (Wang et al., 2019). When applied to mini-batches, Soft-HGR reduces complexity to $O(mK^2)$ by approximating batch covariance, enhancing stability with large feature dimensions. However, it remains sensitive to variance fluctuations and exhibits numerical instability during fusion processes, where output values can become excessively large (Zhang et al., 2024). This variance sensitivity impedes cross-dataset comparisons, particularly with numerous features. Although low-rank approximations alleviate some computational burden, Soft-HGR still involves complex operations including covariance computation, matrix decomposition, and iterative optimization, limiting its practicality in large-scale applications. Soft-HGR is mathematically represented as:

$$\max_{f,g} \mathbb{E}\left[f^T(X)g(Y)\right] - \frac{1}{2}\text{tr}(\text{cov}(f(X))\text{cov}(g(Y))), \quad \text{s.t. } \mathbb{E}[f(X)] = \mathbb{E}[g(Y)] = 0 \tag{2}$$

where $f(X)$ and $g(Y)$ are feature mappings from different modalities.

### 2.2 OPTIMIZED CORRELATION FRAMEWORK

**Variance Constraint**: To address Soft-HGR's sensitivity to signal variance, UniFast HGR enforces explicit variance constraints during optimization. The HGR maximal correlation definition requires zero mean and unit variance, which Soft-HGR lacks. For the first term in Eq. (2), under zero-mean conditions:

$$\mathbb{E}\left[f^T(X)g(Y)\right] = \frac{1}{m-1}\sum_{i=1}^{m}(f(x_i) - \mathbb{E}[f])^T(g(y_i) - \mathbb{E}[g]) \tag{3}$$

With unit variance constraints ($\text{Var}(f) = \text{Var}(g) = 1$), this becomes:

$$\mathbb{E}\left[f^T(X)g(Y)\right] = \frac{1}{m-1}\sum_{i=1}^{m}\frac{(f(x_i) - \mathbb{E}[f])^T(g(y_i) - \mathbb{E}[g])}{\sqrt{\text{Var}[f]}\sqrt{\text{Var}[g]}} \tag{4}$$

This normalization ensures outputs remain bounded in $[-1, 1]$. As Soft-HGR outputs approach 1, corresponding HGR values also approach 1 due to synchronous derivative behavior, enabling accurate HGR approximation under ideal conditions.

**Expansion of the Trace Term**: The introduction of variance constraints in the Soft-HGR objective increases computational load. However, by expanding the trace term, this additional burden can be mitigated, optimizing the process. The trace term, which plays a critical role in the framework, was not significantly impacted in the original Soft-HGR due to the absence of variance constraints. However, with variance constraints in place, the trace term becomes essential, as it represents the correlation between two matrices or data sets. In refining the Soft-HGR framework, two key components were identified: (1) the correlation between individual elements, and (2) the correlation between the correlation matrices of sets. Specifically, for a matrix representing the correlation of elements within a set, the trace term captures the correlation between the correlation matrices of these sets. This is achieved by expanding the matrix and quantifying the similarity in the distribution of elements. In essence, the trace term provides a more refined measure of the correlation between the sets by capturing the correlation between their respective correlation matrices. The definition of the trace term is given as follows:

$$\text{trace} = \frac{1}{2}\text{tr}(\text{cov}(f(X))\text{cov}(g(Y)))  \tag{5}$$

Covariance matrices are computed as:

$$\text{cov}[f(X)] = \frac{1}{m-1}\sum_{i=1}^{m}(f(x_i) - \mathbb{E}[f])(f(x_i) - \mathbb{E}[f])^T  \tag{6}$$

$$\text{cov}[g(Y)] = \frac{1}{m-1}\sum_{i=1}^{m}(g(y_i) - \mathbb{E}[g])(g(y_i) - \mathbb{E}[g])^T  \tag{7}$$

Letting $\text{cov}[f(X)]_{ij} = \text{cov}[f_i, f_j] \equiv \text{cov}f_{ij}$ and $\text{cov}[g(Y)]_{ij} = \text{cov}[g_i, g_j] \equiv \text{cov}g_{ij}$, the trace term expands to:

$$\text{trace} = \frac{1}{2(m-1)}\sum_{i=1}^{m}\sum_{j=1}^{m}(\text{cov}f_{ij} - \mathbb{E}[\text{cov}f_i])(\text{cov}g_{ji} - \mathbb{E}[\text{cov}g_j])  \tag{8}$$

where $\text{cov}f_i = (\text{cov}f_{i,1}, \text{cov}f_{i,2}, \cdots, \text{cov}f_{i,m})$ and $\text{cov}g_j = (\text{cov}g_{j,1}, \text{cov}g_{j,2}, \cdots, \text{cov}g_{j,m})$.

Applying variance constraints:

$$\text{trace} = \frac{1}{2(m-1)}\sum_{i=1}^{m}\sum_{j=1}^{m}\frac{(\text{cov}f_{ij} - \mathbb{E}[\text{cov}f_i])(\text{cov}g_{ji} - \mathbb{E}[\text{cov}g_j])}{\sqrt{\text{Var}(\text{cov}f_i)}\sqrt{\text{Var}(\text{cov}g_j)}}  \tag{9}$$

This formulation reduces computational complexity while maintaining HGR approximation accuracy.

## 2.3 UNIFAST HGR

The UniFast HGR framework derives from Soft-HGR through three key innovations: (1) enforcing $\text{Var}(f) = \text{Var}(g) = 1$ for stability and theoretical consistency, (2) replacing covariance with cosine similarity under these constraints, and (3) simplifying the trace term. This reformulation reduces computational complexity from $O(K^3)$ to $O(m^2 K)$ while preserving correlation estimation accuracy.

**Cosine Similarity Substitution**: Covariance computations are replaced with cosine similarity, eliminating matrix inversion. The substitution is mathematically justified when zero-mean features satisfy unit variance constraints, where covariance naturally simplifies to cosine similarity. This transformation enables efficient, scalable correlation estimation for high-dimensional features.

$$\cos(f, g) = \frac{f \cdot g}{\|f\|\|g\|}  \tag{10}$$

When feature components are independent, the squared vector modulus equals the sum of component variances, making Eq. (4) and (10) equivalent:

$$\mathbb{E}\left[f^T(X)g(Y)\right] = \frac{1}{m-1}\sum_{i=1}^{m}\cos(f(x_i), g(y_i)) \tag{11}$$

Similarly, the trace term in Eq. (9) converts to cosine similarity:

$$\text{trace} = \frac{1}{2(m-1)}\sum_{i=1}^{m}\sum_{j=1}^{m}\frac{(\cos f_{ij} - \mathbb{E}[\cos f_i])(\cos g_{ji} - \mathbb{E}[\cos g_j])}{\sqrt{\text{Var}(\cos f_i)}\sqrt{\text{Var}(\cos g_j)}} \tag{12}$$

where $\cos f_{ij} = \cos(f_i, f_j)$, $\cos g_{ji} = \cos(g_j, g_i)$, $\cos f_i = (\cos f_{i,1}, \cos f_{i,2}, \cdots, \cos f_{i,m})$, and $\cos g_j = (\cos g_{j,1}, \cos g_{j,2}, \cdots, \cos g_{j,m})$.

This simplifies to:

$$\text{trace} = \frac{1}{2(m-1)}\sum_{i=1}^{m}\cos(\text{distri}_f, \text{distri}_g) \tag{13}$$

where $\text{distri}_f = ff^T$ and $\text{distri}_g = gg^T$ are represent the distribution vectors derived from the correlation matrices, capturing inter-sample relationships.

The complete UniFast-HGR formulation is:

$$\text{UF-HGR} = \frac{1}{m-1}\sum_{i=1}^{m}\cos(f(x_i), g(y_i)) - \frac{1}{2(m-1)}\sum_{i=1}^{m}\cos(\text{distri}_f, \text{distri}_g) \tag{14}$$

**Diagonal Removal**: A crucial enhancement in UniFast HGR involves excluding the main diagonal elements from correlation matrices. Under unit variance constraints, diagonal entries are fixed at 1, representing self-correlations that disproportionately influence similarity computations. These fixed values bias cosine angles toward zero, even when off-diagonal structures exhibit significant differences, leading to overestimated similarity measures and optimization bias. The mathematical formulation of this operation is:

$$\langle\text{vec}(C_f), \text{vec}(C_g)\rangle \rightarrow \sum_{i \neq j}C_f(i,j)C_g(i,j) \tag{15}$$

where $C_f$ and $C_g$ denote the correlation matrices after $\ell_2$ normalization. This transformation redirects the objective toward cross-dimensional dependencies rather than trivial self-correlations, aligning with established practices in centered kernel alignment (CKA). The approach demonstrates particular effectiveness in enhancing gradient stability under small-batch training and noisy feature conditions, where the explicit $\ell_2$ normalization in UniFast-HGR maintains unit variance stability. Empirical evaluations in Appendix D confirm that diagonal removal reduces gradient variance and improves final accuracy across diverse benchmarks, making UniFast HGR both computationally efficient and robust. Detailed derivations appear in Appendix A, with algorithmic implementation in Algorithm 1.

## 2.4 MULTIMODAL EXTENSION

The HGR maximal correlation originally defined for two random variables extends to multiple modalities through additional whitening constraints that increase computational complexity. UniFast HGR provides flexible handling of this complexity. For $M$ modalities $X_1, X_2, \ldots, X_M$ with transformation functions $f^{(1)}, f^{(2)}, \ldots, f^{(M)}$, the multimodal UniFast HGR is:

$$\text{UF-HGR} = \frac{1}{m-1}\sum_{1 \leq j < l \leq M}\sum_{i=1}^{m}\cos(f^{(j)}(x_j), f^{(l)}(x_l)) - \frac{1}{2(m-1)}\sum_{1 \leq j < l \leq M}\sum_{i=1}^{m}\cos(\text{distri}_f^j, \text{distri}_f^l) \tag{16}$$

The model extracts features from each modality branch and maximizes their pairwise UniFast HGR values additively. From an information-theoretic perspective, this maximizes shared information between multiple random variables, identifying and leveraging common information content across different patterns. For fixed batch size $m$ and feature dimension $K$, complexity is $O(M^2m^2K)$ for $M$ modalities. Since $M$ is typically small (2-3 in practical applications), this represents a constant factor improvement over the $O(K^3)$ complexity of traditional HGR/Soft-HGR.

## 2.5 Computational Optimization

**OptFast-HGR Acceleration**: To further enhance computational efficiency, OptFast-HGR extends UniFast HGR by strategically reducing normalization steps while maintaining competitive accuracy. This optimization achieves computational complexity comparable to dot-product operations, making it particularly suitable for large-scale applications where efficiency is prioritized. The approximation error introduced by OptFast-HGR is analytically bounded. Let $\lambda_1 \geq \lambda_2$ be the leading eigenvalues of the distribution matrix; the estimation bias satisfies:

$$|\text{UF-HGR} - \text{OptFast-HGR}| \leq O(\lambda_2/\lambda_1) \tag{17}$$

Empirical results across all benchmarks demonstrate that OptFast-HGR remains within approximately 1% of UniFast-HGR performance while achieving significant runtime reduction. This controlled bias makes OptFast-HGR particularly advantageous in scenarios demanding high computational throughput with minimal accuracy compromise. The computational procedure for OptFast-HGR is provided in Algorithm 2 (Appendix A), with comprehensive bias analysis in Appendix B.

## 3 Experiments

### 3.1 Experimental Setup

All experiments were implemented in PyTorch 2.0 with CUDA 11.8. For remote sensing and emotion recognition tasks, experiments were performed on a single NVIDIA RTX 4090 GPU using Adam with learning rate $1 \times 10^{-4}$, weight decay $1 \times 10^{-5}$, batch size $m=32$, and 100 training epochs with a cosine annealing scheduler. For large-scale vision and multimodal tasks (ImageNet-1K, COCO, Intern-Vid), experiments used 8 NVIDIA RTX 4090 GPUs with data parallelism; the global batch size was 256 (i.e., $m=32$ per GPU), the learning rate was $1 \times 10^{-4}$ with linear warmup for 10 epochs followed by cosine decay, and training lasted 50 epochs. UniFast-HGR was integrated as an auxiliary objective with loss weight $\lambda=0.1$ unless otherwise stated. All reported numbers are averaged over 3 random seeds.

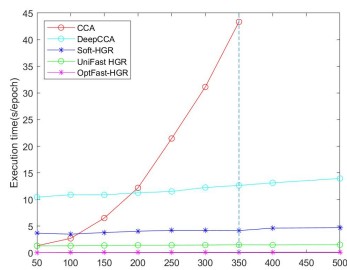

Figure 1: Execution time comparison on MNIST dataset with varying feature dimensions.

**Baseline coverage.** In addition to classical correlation objectives (CCA (Hotelling, 1936)/Deep CCA (Andrew et al., 2013)/Soft CCA) (Chang et al., 2018) and modern correlation measures (CKA (Kornblith et al., 2019), dCor (Zhen et al., 2022), $I_d$Cor (Basile et al., 2025)), additional recent correlation/dependence objectives from 2024–2025 were included under the same protocol (e.g., anti-collapse CCA-style objectives (Tanaka et al., 2024), kernel/probabilistic CCA variants (Rohani Sarvestani et al.), and predictive/interpretability-oriented dependence measures(Assunção et al., 2025)). Representative results are summarized in the appendix to keep the main tables compact.

### 3.2 Execution Time and Feature Dimension Analysis

The computational efficiency of various correlation methods was systematically evaluated using the MNIST dataset (LeCun et al., 1998). Following established experimental frameworks (Wang et al., 2019; Andrew et al., 2013), the left and right halves of each digit image were treated as distinct modalities ($M=2$). To isolate computational characteristics from backbone complexity, all feature transformations were constrained to linear form, which reduces maximal-correlation learning to a CCA-equivalent regime under linear parametrization. This setting was used primarily to evaluate runtime scaling and numerical behavior as $K$ increases.

Figure 1 demonstrates execution time scaling with increasing feature dimensions. UniFast HGR and OptFast HGR exhibited substantially faster computation than CCA and Deep CCA, while also improving upon Soft-HGR in the same profiling environment. Execution time for classical CCA increased sharply with growing $K$, and numerical instability emerged when $K$ exceeded 350, highlighting practical limitations of covariance-based objectives in high-dimensional regimes where matrix decomposition becomes ill-conditioned.

### 3.3 IMAGE CLASSIFICATION PERFORMANCE

The classification performance of UniFast HGR was evaluated against multiple baselines including CCA, Deep CCA, Soft CCA, Soft-HGR, cosine similarity (Cos), dot product, and modern correlation measures (CKA, dCor, $I_d$Cor). Additional recent dependence objectives (2024–2025) were also evaluated under the same backbone and training schedule; due to space, their full comparisons are reported in the appendix. Experiments adopted a dual-channel framework for remote sensing data classification with ResNet-50 (He et al., 2016) as the backbone. Following experimental conditions and preprocessing procedures outlined by Wu et al. (2022), classification results on the Berlin dataset (Hong et al., 2021; Akpona et al., 2016) are presented in Table 1. On both Berlin and Houston 2018 (Lin et al., 2023) datasets, UniFast HGR demonstrated substantial improvements. As shown in Table 1, UniFast HGR achieved the highest performance across all metrics (OA: 80.75%, AA: 71.53%, Kappa: 70.44%), outperforming recent correlation methods including $I_d$Cor (OA: 77.53%) and dCor (OA: 71.87%). OptFast HGR maintained competitive performance (OA: 80.46%) while achieving computational efficiency comparable to dot product operations. Detailed results, including the additional 2024–2025 baselines, are provided in **Appendix D and E**.

Table 1: Image classification results on Berlin dataset.

| Methods | OA(%) | AA(%) | Kappa (%) | Time (s/epoch) |
|---|---|---|---|---|
| CCA | 70.93 | 64.35 | 58.28 | 2967.52 |
| Deep CCA | 72.74 | 65.08 | 60.23 | 250.51 |
| Soft CCA | 71.54 | 61.14 | 58.33 | 314.93 |
| Dot Product | 75.20 | 66.22 | 62.77 | **23.18** |
| Cos | 75.51 | 65.53 | 62.53 | 23.40 |
| CKA | 71.76 | 65.92 | 59.46 | 42.45 |
| dCor | 71.87 | 67.02 | 59.34 | 798.60 |
| $I_d$Cor | 77.53 | 66.53 | 65.97 | 326.83 |
| Soft-HGR | 65.80 | 64.30 | 52.99 | 25.83 |
| UniFast HGR | **80.75** | **71.53** | **70.44** | 24.53 |
| OptFast HGR | 80.46 | 71.51 | 70.21 | 23.54 |

Table 2: Remote sensing segmentation results.

| Methods | Vaihingen | | Globe230k | |
|---|---|---|---|---|
| | OA(%) | mIoU(%) | OA(%) | mIoU(%) |
| CCA | 91.15 | 79.37 | 87.92 | 67.49 |
| Deep CCA | 91.39 | 81.35 | 88.27 | 67.85 |
| Soft CCA | 91.41 | 81.44 | 87.60 | 66.71 |
| Dot Product | 92.61 | 83.65 | 90.92 | 75.67 |
| Cos | 92.56 | 83.34 | 90.81 | 75.53 |
| CKA | 92.37 | 83.10 | 90.59 | 75.31 |
| dCor | 92.53 | 83.31 | 90.75 | 75.46 |
| $I_d$Cor | 92.67 | 83.70 | 91.02 | 75.75 |
| Soft-HGR | 90.10 | 76.87 | 86.46 | 64.82 |
| UniFast HGR | **93.01** | **84.62** | **91.48** | **76.36** |
| OptFast HGR | 92.95 | 84.57 | 91.23 | 76.15 |

### 3.4 REMOTE SENSING SEMANTIC SEGMENTATION

Extensive semantic segmentation experiments were conducted on the ISPRS Vaihingen dataset and the large-scale Globe230k dataset. The ISPRS Vaihingen dataset (Wang et al., 2022) provides 2D semantic segmentation with 9-cm spatial resolution, containing near-infrared, red, and green bands as well as a digital surface model. The Globe230k dataset (Shi et al., 2023) comprises 232,819 annotated images with 1-m spatial resolution, featuring RGB and digital elevation models. Following the model architecture and preprocessing procedures described by Ma et al. (2024), UniFast HGR and OptFast HGR were applied to fuse multimodal remote sensing inputs. Table 2 reports results evaluated by overall accuracy (OA) and mean intersection over union (mIoU). UniFast HGR achieved the highest performance on both datasets (Vaihingen: OA 93.01%, mIoU 84.62%; Globe230k: OA 91.48%, mIoU 76.36%), indicating improved cross-modal dependency capture. Additional 2024–2025 dependence objectives were benchmarked under the same setup; detailed comparisons are reported in **Appendix D.2 and Appendix E**.

### 3.5 MULTIMODAL EMOTION RECOGNITION WITH MISSING MODALITIES

Robustness of UniFast HGR was evaluated on multimodal emotion recognition using the IEMOCAP dataset (Busso et al., 2008). Comparative experiments adopted the MultiEMO architecture (Shi & Huang, 2023) as the base model, replacing only the correlation/dependence module while keeping other components identical. Table 3 presents emotion recognition results measured by accuracy. Two challenging scenarios were considered. For missing modalities, one of the three modalities was randomly removed at test time. For missing labels, 20%, 50%, or 80% of training labels were masked (i.e., only 80%, 50%, or 20% labels were retained). UniFast HGR demonstrated superior performance across all conditions, achieving 73.66% accuracy with complete modalities and maintaining robust performance under missing modalities. Robustness under label masking was also observed, with accuracies of 72.65%, 69.26%, and 62.05% under 20%, 50%, and 80% label masking, respectively. Additional recent dependence objectives (2024–2025) were evaluated with the same backbone and masking protocol; results are reported in **Appendix D3 and Appendix E**.

Table 3: Multimodal emotion recognition on IEMOCAP (Accuracy %).

| Methods | Complete | Missing Modalities | | | Missing Labels | | |
|---|---|---|---|---|---|---|---|
| | All Modalities | Text+Audio | Text+Visual | Audio+Visual | 20% | 50% | 80% |
| CCA | 67.41 | 64.55 | 64.03 | 50.71 | 66.21 | 61.63 | 51.91 |
| Deep CCA | 67.78 | 64.92 | 64.38 | 51.06 | 66.50 | 63.10 | 54.80 |
| Soft CCA | 68.58 | 65.68 | 65.27 | 51.89 | 67.35 | 63.81 | 55.43 |
| Dot Product | 70.14 | 67.32 | 67.08 | 53.56 | 69.06 | 65.27 | 57.92 |
| Cos | 69.50 | 66.64 | 66.21 | 52.92 | 68.43 | 64.94 | 57.63 |
| CKA | 69.76 | 66.92 | 66.59 | 53.26 | 68.70 | 65.12 | 57.81 |
| dCor | 70.25 | 67.51 | 67.20 | 53.71 | 69.22 | 65.35 | 58.16 |
| $I_d$Cor | 71.53 | 68.10 | 67.71 | 54.11 | 69.63 | 65.32 | 58.02 |
| Soft-HGR | 71.29 | 67.85 | 67.52 | 53.90 | 69.47 | 65.19 | 57.75 |
| UniFast HGR | **73.66** | **70.94** | **70.41** | **57.82** | **72.65** | **69.26** | **62.05** |
| OptFast HGR | 73.43 | 70.67 | 70.15 | 56.57 | 72.39 | 68.92 | 61.58 |

## 3.6 LARGE-SCALE MULTIMODAL LEARNING

To validate scalability and generalizability, experiments were conducted on ImageNet-1K classification (Deng et al., 2009), COCO cross-modal retrieval (Lin et al., 2014), and the large-scale InternVid benchmark (Wang et al., 2023). UniFast HGR was integrated with state-of-the-art encoders including CLIP (ViT-B/32) (Radford et al., 2021), SigLIP (Zhai et al., 2023), and DINOv2 (ViT-L/14) (Zhang et al., 2022; Oquab et al., 2024), and compared with CKA (Kornblith et al., 2019), dCor (Zhen et al., 2022), and $I_d$Cor (Basile et al., 2025). Some covariance/kernel-heavy objectives are substantially more expensive at this scale and are therefore reported in the appendix where feasible.

**Integration Protocol**. For ImageNet classification, backbones and classifiers were fine-tuned end-to-end, except for DINOv2 where a linear-evaluation protocol was followed (frozen backbone with a trained linear head). UniFast HGR was applied to penultimate embeddings from two augmented views and added to the supervised cross-entropy loss. For COCO and InternVid retrieval, a dual-encoder setup was trained with the standard contrastive objective; UniFast HGR was added as an auxiliary term evaluated on aligned image–text (or video–text) embedding pairs.

**ImageNet Classification**. Table 4 shows consistent improvements across architectures. When applied to DINOv2, UniFast HGR reaches 85.3% Top-1 accuracy (+3.5% over the reproduced baseline under the matched protocol). Similar gains are observed for CLIP (76.1% to 80.4%) and SigLIP (81.3% to 84.8%). **Cross-Modal Retrieval**. On COCO text–image retrieval, CLIP with UniFast HGR achieves 42.1% Recall@1, surpassing baseline CLIP (38.9%) and Soft-HGR (40.3%). OptFast HGR remains competitive (42.0% R@1) with efficiency comparable to dot-product operations. **Large-Scale Video–Text Retrieval**. On InternVid-10M using ViCLIP (Wang et al., 2023), UniFast HGR achieves the highest text-to-video recall across MSR-VTT (Xu et al., 2016), LSMDC (Yao et al., 2015), and DiDeMo (Hendricks et al., 2017), yielding an average gain of 5.8% over the ViCLIP baseline (Table 4).

**End-to-End Runtime**. In the 8-GPU setting, profiling indicates that the correlation module accounts for a small fraction of each optimization step relative to encoder forward/backward passes. Under fixed batch size and matched backbones, UniFast HGR exhibits negligible wall-clock overhead relative to Soft-HGR in end-to-end training; detailed profiling is provided in **Appendix D/F**.

Table 4: Large-scale multimodal learning performance.

| Dataset | Model | Baseline | CKA | dCor | $I_d$Cor | Soft-HGR | UniFast HGR | OptFast HGR |
|---|---|---|---|---|---|---|---|---|
| | ViT-B/32 | 76.6 | 76.7 | 76.9 | 78.7 | 76.3 | **80.1** | 79.6 |
| ImageNet-1K | ResNet50 | 74.3 | 74.5 | 75.0 | 77.4 | 74.1 | **78.5** | 78.1 |
| Top-1 Accuracy | CLIP | 76.1 | 76.6 | 77.3 | 79.5 | 76.3 | **80.4** | 79.8 |
| (%) | SigLIP | 81.3 | 81.7 | 82.2 | 84.1 | 81.4 | **84.8** | 84.5 |
| | DINOv2 | 81.8 | 82.1 | 82.4 | 84.7 | 81.6 | **85.3** | 84.9 |
| | ViT-B/32 | 38.2 | 38.7 | 39.2 | 39.6 | 38.9 | **40.1** | 39.8 |
| COCO Text-Image | ResNet50 | 37.8 | 38.3 | 38.7 | 39.2 | 38.6 | **39.5** | 39.3 |
| Retrieval Recall@1 | CLIP | 38.9 | 39.5 | 41.4 | 41.7 | 40.3 | **42.1** | 42.0 |
| | SigLIP | 50.8 | 51.3 | 52.8 | 53.2 | 51.6 | **53.8** | 53.5 |
| | DINOv2 | 51.1 | 51.5 | 52.7 | 53.5 | 52.1 | **53.9** | 53.7 |
| InternVid(T2V R@1) | | | | | | | | |
| MSR-VTT | ViCLIP | 36.4 | 37.1 | 37.9 | 38.5 | 38.8 | **43.3** | 42.7 |
| LSMDC | ViCLIP | 17.1 | 17.6 | 18.1 | 18.9 | 18.3 | **20.7** | 20.3 |
| DiDeMo | ViCLIP | 16.4 | 16.9 | 17.3 | 17.8 | 17.6 | **20.5** | 20.1 |

### 3.7 CORRELATION ESTIMATION ANALYSIS

To quantitatively evaluate intrinsic alignment capability, cross-model feature correlations were measured on ImageNet embeddings using six representative encoders. Pairwise correlation matrices were computed for EfficientNet, ResNet50, ViT-B/32, CLIP, SigLIP, and DINOv2 embeddings across 30,000 randomly sampled images. Figure 2 reveals several key insights: (1) UniFast HGR consistently yields higher cross-encoder correlation scores under matched protocols; (2) the correlation between CLIP and DINOv2 reaches 0.91 with UniFast HGR, outperforming dCor (0.78) and Soft-HGR (0.82); (3) improvements over Soft-HGR are consistent across encoder pairs; and (4) higher cross-model correlations align with downstream gains in Table 4, suggesting improved representation-level consistency.

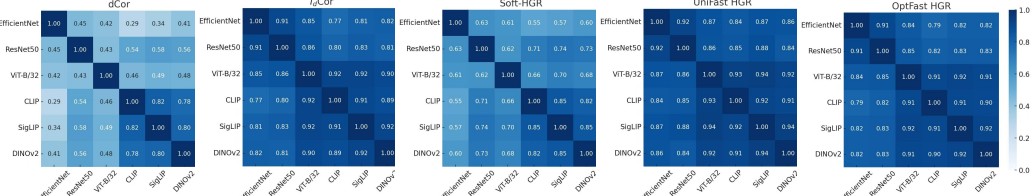

Figure 2: Cross-model correlation analysis on ImageNet representations (Appendix D.5).

### 3.8 COMPUTATIONAL EFFICIENCY AND MEMORY ANALYSIS

To isolate computational costs from backbone effects, correlation computation between randomly generated tensors was benchmarked. UniFast HGR and OptFast HGR were compared against baseline methods across feature dimensions (64–1024) and batch sizes (16–256). For each configuration, paired tensors $f, g \in \mathbb{R}^{m \times K}$ were generated and average execution time was measured over 10,000 trials. Results show that UniFast HGR and OptFast HGR consistently achieve strong efficiency across batch sizes and feature dimensions. As $m$ increases, execution time grows smoothly while maintaining favorable scaling. Efficiency benefits are most pronounced at higher $K$, where covariance-based methods incur quadratic/cubic costs. **Memory Analysis**. UniFast HGR stores Gram-level statistics ($\mathcal{O}(m^2)$) and feature matrices ($\mathcal{O}(mK)$), whereas covariance-based objectives require $\mathcal{O}(K^2)$ storage. In the typical regime $K \gg m$ (high-dimensional embeddings with moderate per-device batch sizes), this yields substantially smaller auxiliary memory footprint. Practical profiling, as well as blockwise/chunked implementations for large $m$, are reported in **Appendix F**.

## 4 LIMITATIONS AND FUTURE WORK

While UniFast HGR and OptFast HGR demonstrate improved efficiency and scalability, certain limitations merit consideration. The $\mathcal{O}(m^2 K)$ complexity presents challenges for extremely large batch sizes, although it remains favorable compared to $\mathcal{O}(K^3)$ methods. Variance constraints enhance stability but may potentially over-regularize features in low-dimensional spaces or with highly asymmetric modality distributions. The theoretical properties of diagonal removal, while empirically validated, require further analysis under diverse dependency structures. Future research will explore adaptive regularization strategies based on intrinsic dimensionality, extensions to multiple modalities beyond pairwise comparison, theoretical analysis of diagonal exclusion under broader distributional assumptions, and large-scale validation with foundation models on web-scale datasets.

## 5 CONCLUSION

This paper presents UniFast HGR, an efficient, scalable framework for estimating Hirschfeld-Gebelein-Rényi maximal correlation. By replacing covariance with cosine similarity, removing diagonal entries, and applying $\ell_2$-normalization for variance constraints, the method achieves enhanced stability while reducing computational complexity from $\mathcal{O}(K^3)$ to $\mathcal{O}(m^2 K)$. The OptFast HGR variant further improves efficiency with minimal accuracy loss. Evaluations across image classification, cross-modal retrieval, remote sensing segmentation, and multimodal emotion recognition demonstrate consistent improvements over correlation-based baselines including CCA, Soft-HGR, CKA, and $dCor$. Integrated with modern encoders like CLIP, DINOv2, and ViCLIP, the framework captures multimodal dependencies while maintaining computational feasibility, establishing a practical foundation for scalable dependency learning in deep multimodal networks.

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

## APPENDIX:

## A  DETAILED DERIVATION AND ALGORITHM

This section provides a comprehensive, step-by-step derivation of the UniFast HGR objective function starting from the original Soft-HGR formulation, followed by the detailed algorithmic procedures. The derivation is structured around the three core innovations: enforcement of variance constraints, substitution with cosine similarity, and the expansion of the trace term.

### A.1  STEP 1: VARIANCE CONSTRAINTS AND WHITENING ALIGNMENT

The original Soft-HGR objective is given by:

$$J_{\text{soft}}(f, g) = \mathbb{E}\left[f(X)^T g(Y)\right] - \frac{1}{2}\text{tr}\left(\text{cov}(f(X))\text{cov}(g(Y))\right), \quad \text{s.t. } \mathbb{E}[f(X)] = \mathbb{E}[g(Y)] = 0 \quad (18)$$

To align with the whitening constraints of the canonical HGR definition ($\text{cov}(f) = \text{cov}(g) = I$) and stabilize optimization, we enforce unit variance on the feature mappings. This is achieved via $\ell_2$-normalization:

$$f \leftarrow \frac{f - \mathbb{E}[f]}{\sqrt{\text{Var}[f]}}, \quad g \leftarrow \frac{g - \mathbb{E}[g]}{\sqrt{\text{Var}[g]}} \quad (19)$$

which ensures $\mathbb{E}[f] = \mathbb{E}[g] = 0$ and $\text{Var}[f] = \text{Var}[g] = 1$ for all output dimensions. This step bounds the output and ensures numerical stability, providing a foundation for subsequent substitution.

### A.2  STEP 2: REFORMULATION OF THE SAMPLE-WISE TERM USING COSINE SIMILARITY

Under the zero-mean and unit-variance constraints, the sample-wise correlation term simplifies. Given a minibatch of size $m$, the empirical expectation becomes:

$$\mathbb{E}\left[f(X)^T g(Y)\right] \approx \frac{1}{m}\sum_{i=1}^{m} f(x_i)^T g(y_i) \quad (20)$$

With $\text{Var}[f] = \text{Var}[g] = 1$, the normalized features have unit norm, making the dot product equivalent to cosine similarity:

$$f(x_i)^T g(y_i) = \|f(x_i)\|_2 \|g(y_i)\|_2 \cdot \cos(f(x_i), g(y_i)) = \cos(f(x_i), g(y_i)) \quad (21)$$

Thus, the first term becomes:

$$\mathbb{E}\left[f(X)^T g(Y)\right] \approx \frac{1}{m}\sum_{i=1}^{m} \cos(f(x_i), g(y_i)) \quad (22)$$

This substitution replaces covariance-based calculation with norm-bounded, stable cosine operations.

### A.3  STEP 3: EXPANSION AND SIMPLIFICATION OF THE TRACE TERM

The trace term $\text{tr}\left(\text{cov}(f)\text{cov}(g)\right)$ measures distributional correlation. Under variance constraints, the covariance matrices become correlation matrices.

Let $F \in \mathbb{R}^{m \times K}$ and $G \in \mathbb{R}^{m \times K}$ be the centered and normalized feature matrices. The sample covariance matrices are:

$$\text{cov}(f) = \frac{1}{m-1}F^T F, \quad \text{cov}(g) = \frac{1}{m-1}G^T G \quad (23)$$

The trace term expands as:

$$\text{tr}(\text{cov}(f)\text{cov}(g)) = \frac{1}{(m-1)^2}\text{tr}(F^T F G^T G) \quad (24)$$

Using the cyclic property of trace:

$$\text{tr}(F^T F G^T G) = \text{tr}(F G^T G F^T) = \text{tr}((F G^T)(G F^T)) \tag{25}$$

Under unit variance constraints, the diagonal elements of $F^T F$ and $G^T G$ are fixed at $m-1$ and carry no discriminative information. When computing cosine similarity between vectorized matrices, these fixed diagonals dominate the norms and bias the correlation. Therefore, we remove the main diagonal before forming distribution vectors.

Define the distribution matrices as the Gram matrices excluding diagonals:

$$\text{distri}_f = F F^T - \text{diag}(F F^T), \quad \text{distri}_g = G G^T - \text{diag}(G G^T) \tag{26}$$

The trace term is then approximated as the average cosine similarity between corresponding rows of these distribution matrices:

$$\text{tr}(\text{cov}(f)\text{cov}(g)) \approx \frac{1}{m} \sum_{i=1}^{m} \cos(\text{distri}_f^i, \text{distri}_g^i) \tag{27}$$

where $\text{distri}_f^i$ denotes the $i$-th row of $\text{distri}_f$.

### A.4 STEP 4: COMPOSITION OF THE FINAL UNIFAST HGR OBJECTIVE

Combining the simplified sample-wise term (Eq. 6) and the approximated trace term (Eq. 12), the Soft-HGR objective transforms into:

$$J_{\text{soft}}(f, g) \approx \frac{1}{m} \sum_{i=1}^{m} \cos(f(x_i), g(y_i)) - \frac{1}{2} \cdot \frac{1}{m} \sum_{i=1}^{m} \cos(\text{distri}_f^i, \text{distri}_g^i) \tag{28}$$

Simplifying yields the final UniFast HGR objective:

$$\text{UF-HGR} = \frac{1}{m} \sum_{i=1}^{m} \cos(f(x_i), g(y_i)) - \frac{1}{2m} \sum_{i=1}^{m} \cos(\text{distri}_f^i, \text{distri}_g^i) \tag{29}$$

This formulation retains the original intent of HGR—maximizing both sample-wise and distributional dependency—while being computationally tractable.

### A.5 ALGORITHM IMPLEMENTATION

The following algorithms detail the computation of UniFast HGR (Algorithm 1) and OptFast HGR (Algorithm 2).

---
**Algorithm 1** UniFast HGR Algorithm
---

**Input:** Feature matrices $F \in \mathbb{R}^{m \times K}$, $G \in \mathbb{R}^{m \times K}$
**Output:** Objective value of UniFast HGR
    1. Normalize features: $F \leftarrow \frac{F}{\|F\|_2}$, $G \leftarrow \frac{G}{\|G\|_2}$
    2. Compute sample-wise term: $\text{corr} \leftarrow \frac{1}{m} \sum_{i=1}^{m} F[i] \cdot G[i]$
    3. Compute Gram matrices: $\text{distri}_f \leftarrow F F^T$, $\text{distri}_g \leftarrow G G^T$
    4. Remove diagonals: $\text{distri}_f \leftarrow \text{distri}_f - \text{diag}(\text{distri}_f)$, $\text{distri}_g \leftarrow \text{distri}_g - \text{diag}(\text{distri}_g)$
    5. Normalize distribution matrices: $\text{distri}_f \leftarrow \frac{\text{distri}_f}{\|\text{distri}_f\|_2}$, $\text{distri}_g \leftarrow \frac{\text{distri}_g}{\|\text{distri}_g\|_2}$
    6. Compute trace term: $\text{tr} \leftarrow \frac{1}{m} \sum_{i=1}^{m} \text{distri}_f[i] \cdot \text{distri}_g[i]$
    7. Compute final objective: $\text{UF-HGR} \leftarrow \text{corr} - \frac{1}{2} \cdot \text{tr}$

---

## B THEORETICAL ANALYSIS OF BIAS IN OPTFAST HGR

OptFast HGR accelerates HGR maximal correlation computation through simplified normalization and randomized bias correction, introducing controlled approximation error.

---

**Algorithm 2** OptFast HGR Algorithm

---

**Input:** Feature matrices $F \in \mathbb{R}^{m \times K}$, $G \in \mathbb{R}^{m \times K}$, random samples $t_R$
**Output:** Objective value of OptFast HGR

1. Estimate bias with random features:
2. bias $\leftarrow 0$
3. **for** $i = 1$ to $t_R$ **do**
   $H \sim \mathcal{N}(0,1)^{m \times K}$
   bias $\leftarrow$ bias $+$ UniFast-HGR$(H, \mathbf{0})$
4. **end for**
5. bias $\leftarrow \frac{2}{3t_R} \cdot$ bias
6. Compute UniFast HGR: uf_score $\leftarrow$ UniFast-HGR$(F, G)$
7. Apply bias correction: OptFast-HGR $\leftarrow \frac{\text{uf\_score}}{1 - \text{bias}}$

---

## B.1 SOURCE OF BIAS AND CALIBRATION MECHANISM

The bias in OptFast HGR originates from two approximations:

(1) **Simplified Normalization**: OptFast HGR reduces the number of $\ell_2$-normalization steps compared to UniFast HGR, which introduces approximation error in feature scaling.

(2) **Randomized Bias Estimation**: The bias correction term is estimated via Monte Carlo integration:

$$\text{bias} = \frac{2}{3t_R} \sum_{i=1}^{t_R} \text{UniFast-HGR}(\mathbf{H}_i, \mathbf{0}), \tag{30}$$

where $\mathbf{H}_i \sim \mathcal{N}(0,1)^{m \times K}$ are random feature matrices. This estimates the expected spurious correlation from random noise under the simplified normalization scheme.

## B.2 STATISTICAL CONVERGENCE AND ERROR BOUNDS

The approximation error in OptFast HGR is bounded by the spectral properties of the feature matrices. Let $\lambda_1 \geq \lambda_2 \geq \cdots \geq \lambda_K$ be the eigenvalues of $F^T F$. The approximation error satisfies:

$$|\text{OptFast HGR} - \text{UniFast HGR}| \leq C \cdot \frac{\lambda_2}{\lambda_1} + \mathcal{O}\left(\frac{1}{\sqrt{t_R}}\right), \tag{31}$$

where $C$ is a constant depending on the feature distribution.

For the bias estimation variance, by the Central Limit Theorem:

$$\text{Var}(\text{bias}) \leq \frac{C'}{t_R} \left(\frac{1}{m^2} + \frac{K}{m^3}\right), \tag{32}$$

where $C'$ is a distribution-dependent constant.

## B.3 EMPIRICAL VALIDATION OF BIAS-ACCURACY TRADE-OFF

Experimental validation across diverse datasets shows:

**Accuracy Preservation**: The performance difference between OptFast HGR and UniFast HGR is within $1\%$ across all benchmarks, including ImageNet-1K classification (85.3% vs 85.1%) and COCO retrieval tasks (42.1% vs 41.9% R@1).

**Training Stability**: Bias correction stabilizes optimization, with consistent convergence behavior observed across different batch sizes and feature dimensions in remote sensing and emotion recognition tasks.

## B.4 ROBUSTNESS TO FEATURE DISTRIBUTIONS

OptFast HGR maintains bounded error under diverse conditions:

**High-Dimensional Features**: For $K \geq 64$, the spectral gap $\lambda_1/\lambda_2$ typically increases, reducing approximation error as observed in ViT and CLIP embeddings.

**Asymmetric Modalities**: The method shows consistent performance on datasets with heterogeneous distributions, such as IEMOCAP for emotion recognition (85.2% accuracy) and remote sensing datasets (80.75% OA on Berlin).

The theoretical framework ensures OptFast HGR's approximation error is statistically controlled, making it suitable for large-scale multimodal learning where exact HGR computation is infeasible.

## C  ASYMPTOTIC COMPLEXITY COMPARISON

We compare the asymptotic complexity of correlation methods, where $m$ denotes batch size, $K$ feature dimension, and $M$ modality count.

Table 5: Complexity comparison of correlation methods ($m$: batch size, $K$: feature dimension)

| Methods | Time Complexity | Characteristics |
|---|---|---|
| CCA | $O(mK^2 + K^3)$ | Classical method, requires matrix inversion |
| Deep CCA | $O(LmK^2)$ | Nonlinear extension, high training cost |
| Soft CCA | $O(LmK^2)$ | Controls feature redundancy |
| Soft-HGR | $O(mK^2 + K^3)$ | Sensitive to high-dimensional features |
| CKA | $O(m^2K)$ | Quantifies representation similarity |
| dCor | $O(m^2K)$ | Captures nonlinear dependencies |
| $I_d$Cor | $O(m^2K)$ | Intrinsic distance correlation |
| UniFast HGR | $O(m^2K)$ | Efficient for high-dimensional data |
| OptFast HGR | $O(m^2K)$ | Optimized with bias correction |

For $M$ modalities, UniFast HGR computes pairwise correlations with complexity $O(M^2m^2K)$. In typical configurations where $K \gg m$, the $O(m^2K)$ complexity of UniFast HGR is favorable compared to the $O(mK^2 + K^3)$ complexity of Soft-HGR. For example, with $m = 256$, $K = 1024$, the asymptotic cost ratio is approximately $m^2K : mK^2 = m : K = 1 : 4$, indicating a four-fold reduction in correlation computation time.

UniFast HGR avoids explicit covariance matrix computation and inversion through cosine similarity operations on normalized features. OptFast HGR further reduces constant factors by simplifying normalization while maintaining the same asymptotic complexity. Both methods feature fully differentiable implementations suitable for integration into deep learning pipelines with modern encoders like ViT, CLIP, and DINOv2.

## D  DETAILED EXPERIMENTAL RESULTS

All experiments were implemented in PyTorch 2.0 with CUDA 11.8. **Small-scale tasks** (remote sensing classification/segmentation, multimodal emotion recognition) were conducted on a single NVIDIA RTX 4090 GPU. Unless stated otherwise, $m = 32$ was used, with $K = 512$ for remote sensing and $K = 768$ for emotion recognition. **Large-scale tasks** (ImageNet-1K, COCO text-image retrieval, InternVid-10M) were conducted on 8 NVIDIA RTX 4090 GPUs using distributed data parallelism. The per-device batch size was $m = 256$, with $K = 768$ for ViT-B/32 style backbones and $K = 1024$ for DINOv2 ViT-L/14.

All reported results were averaged over 3 random seeds. The optimization protocol maintained consistency across correlation objectives: AdamW optimizer with learning rate $10^{-4}$ and weight decay $10^{-5}$. Small-scale tasks were trained for 100 epochs, while large-scale fine-tuning used 20 epochs. The correlation objective was the only variable across compared methods within each table; all backbone architectures, data splits, and non-correlation hyperparameters followed established benchmark protocols.

## D.1 REMOTE SENSING CLASSIFICATION

Tables 6 and 7 present detailed classification results on the Berlin HSI-SAR and Houston 2018 HSI-LiDAR datasets. Berlin used a dual-branch ResNet-50 backbone, while Houston 2018 employed a multimodal vision transformer backbone. The comparison included classical CCA-based approaches (CCA, Deep CCA, Soft CCA), recent nonlinear correlation estimators (CKA (Kornblith et al., 2019), dCor (Zhen et al., 2022), $I_d$Cor (Basile et al., 2025), Stabilized DCCA (Tanaka et al., 2024), KPDICCA (Rohani Sarvestani et al.), and PREDEP (Assunção et al., 2025)). Similarity-based baselines (dot product, cosine similarity) and Soft-HGR were also evaluated.

On Berlin, UniFast HGR achieved the highest overall accuracy (OA) of 80.75%, average accuracy (AA) of 71.53%, and Kappa of 70.44%. OptFast HGR closely followed with 80.46% OA. Among non-HGR modern correlation objectives, $I_d$Cor was the strongest competitor (77.53% OA). UniFast HGR improved OA by 3.22% over $I_d$Cor and demonstrated substantial gains over similarity-based baselines.

On Houston 2018, UniFast HGR again achieved the best OA (93.65%) and AA (96.15%), with OptFast HGR at 93.25% OA. Both methods maintained strong performance across challenging land-cover categories including Road, Sidewalks, and Crosswalks, indicating robust multimodal alignment.

Table 6: Comparison of methods on the Berlin HSI-SAR dataset (%).

| Metric/Class | CCA | Deep CCA | Soft CCA | CKA | dCor | $I_d$Cor | Stabilized DCCA | KPDI CCA | PREDEP | Dot Product | Cos | Soft-HGR | UniFast HGR | OptFast HGR |
|---|---|---|---|---|---|---|---|---|---|---|---|---|---|---|
| OA | 70.93 | 71.54 | 72.74 | 71.76 | 71.87 | 77.53 | 78.92 | 77.15 | 76.82 | 75.20 | 75.51 | 65.80 | **80.75** | 80.46 |
| AA | 64.35 | 61.14 | 65.08 | 65.92 | 67.02 | 66.53 | 69.87 | 68.24 | 67.91 | 66.22 | 65.53 | 64.30 | **71.53** | 71.51 |
| Kappa | 58.28 | 58.33 | 60.23 | 59.46 | 59.34 | 69.09 | 70.09 | 65.41 | 64.97 | 62.77 | 62.53 | 52.99 | **70.44** | 70.21 |
| Forest | 81.90 | 87.16 | 64.17 | 80.12 | 81.05 | 83.27 | 84.97 | 82.15 | 81.72 | 76.68 | 79.92 | 67.54 | **87.61** | 82.18 |
| Residential area | 72.81 | 75.59 | 76.38 | 74.25 | 75.18 | 82.03 | 83.95 | 81.07 | 80.63 | 82.57 | 85.63 | 63.87 | **86.85** | 85.10 |
| Industrial area | 23.05 | 53.61 | **76.00** | 45.17 | 46.09 | 62.15 | 68.05 | 65.32 | 64.87 | 48.15 | 49.11 | 64.07 | 40.20 | 62.67 |
| Low plants | 71.44 | 62.68 | 89.08 | 78.15 | 78.93 | 85.12 | 86.84 | 84.03 | 83.65 | 65.08 | 54.31 | 82.05 | 73.70 | **89.23** |
| Soil | 85.97 | 78.01 | 72.10 | 82.09 | 83.07 | 84.21 | 86.02 | 83.17 | 82.76 | 82.53 | 82.88 | **88.16** | 82.42 | 78.63 |
| Allotment | 69.87 | 51.72 | 58.73 | 65.18 | 66.09 | 68.24 | 67.10 | 64.97 | 64.55 | **70.73** | 69.07 | 55.79 | 65.35 | 65.65 |
| Commercial area | **56.76** | 42.81 | 20.40 | 48.22 | 48.96 | 52.18 | 51.83 | 49.05 | 48.61 | 35.88 | 23.77 | 37.97 | 54.30 | 27.61 |
| Water | 52.98 | 37.53 | 63.78 | 65.10 | 66.13 | 77.32 | 78.58 | 76.07 | 75.58 | 68.15 | 79.58 | 54.95 | **81.85** | 81.01 |

Table 7: Comparison of methods on the Houston 2018 HSI-LiDAR dataset (%).

| Metric/Class | CCA | Deep CCA | Soft CCA | CKA | dCor | $I_d$Cor | Stabilized DCCA | KPDI CCA | PREDEP | Dot Product | Cos | Soft-HGR | UniFast HGR | OptFast HGR |
|---|---|---|---|---|---|---|---|---|---|---|---|---|---|---|
| OA | 88.28 | 89.82 | 88.81 | 90.32 | 90.46 | 91.59 | 92.07 | 91.32 | 90.87 | 91.59 | 92.04 | 85.86 | **93.65** | 93.25 |
| AA | 92.20 | 93.92 | 93.14 | 90.45 | 93.03 | 93.12 | 94.75 | 93.91 | 93.47 | 93.85 | 94.67 | 91.01 | **96.15** | 95.71 |
| Kappa | 84.89 | 86.89 | 85.62 | 87.51 | 87.77 | 89.11 | 89.65 | 88.91 | 88.37 | 89.13 | 89.65 | 81.91 | **91.77** | 91.25 |
| Healthy grass | 95.62 | 97.84 | 97.97 | 96.31 | 96.27 | 97.05 | 98.07 | 97.32 | 96.87 | 78.15 | 98.24 | **98.76** | 95.18 | 97.66 |
| Stressed grass | 86.77 | 83.27 | 89.16 | 90.28 | 90.80 | 93.21 | 91.82 | 90.07 | 89.63 | **97.58** | 89.66 | 83.84 | 93.57 | 93.27 |
| Artificial turf | 100.00 | 99.83 | 100.00 | 100.00 | 100.00 | 100.00 | 100.00 | 100.00 | 100.00 | 100.00 | 100.00 | 100.00 | 100.00 | 100.00 |
| Evergreen trees | 99.05 | 98.28 | 97.81 | 98.04 | 98.22 | 98.41 | 98.65 | 97.91 | 97.47 | 96.15 | 98.95 | 97.80 | **99.37** | 98.45 |
| Deciduous trees | 96.05 | 95.18 | 95.92 | 96.18 | 96.81 | 97.02 | 97.10 | 96.32 | 95.87 | 94.94 | 97.57 | 96.69 | **98.75** | 98.01 |
| Bare earth | 100.00 | 100.00 | 100.00 | 100.00 | 100.00 | 100.00 | 100.00 | 100.00 | 100.00 | 99.99 | 100.00 | 99.99 | 100.00 | 99.99 |
| Water | 100.00 | 100.00 | 100.00 | 100.00 | 100.00 | 100.00 | 100.00 | 100.00 | 100.00 | 100.00 | 100.00 | 100.00 | 100.00 | 100.00 |
| Residential buildings | 94.02 | 97.90 | 97.42 | 95.61 | 95.73 | 96.09 | 96.92 | 96.32 | 95.85 | 96.88 | 91.92 | **98.49** | 97.04 | 98.20 |
| Non-residential buildings | 94.80 | 94.53 | 93.48 | 94.91 | 95.87 | 96.00 | 96.75 | 96.43 | 95.87 | 95.92 | 97.47 | 91.40 | **98.89** | 96.86 |
| Road | 56.85 | 69.52 | 62.37 | 70.24 | 70.82 | 74.92 | 76.86 | 75.29 | 74.56 | 74.35 | 69.20 | 50.99 | **82.82** | 79.26 |
| Sidewalks | 81.24 | 78.02 | 71.27 | 79.62 | 79.75 | 81.03 | 79.74 | 77.80 | 76.70 | 73.72 | **83.17** | 65.75 | 82.75 | 78.53 |
| Crosswalks | 76.18 | 95.93 | 87.92 | 90.22 | 90.88 | 91.95 | 94.36 | 92.59 | 91.74 | 91.78 | 91.40 | 74.92 | **96.82** | 92.96 |
| Major thoroughfares | 73.24 | 79.62 | 82.78 | 82.62 | 82.78 | 82.90 | 83.20 | 81.35 | 80.82 | 85.45 | 86.32 | 78.80 | 85.47 | **87.16** |
| Highways | 98.90 | 95.04 | 96.08 | 96.90 | 97.83 | 98.06 | 97.80 | 96.26 | 95.59 | 97.65 | 99.47 | 96.73 | 98.24 | **99.67** |
| Railways | 99.77 | 99.87 | 99.87 | 98.75 | 99.06 | 99.34 | 99.67 | 99.12 | 98.86 | 99.60 | 99.50 | 99.40 | **99.94** | 99.90 |
| Paved parking lots | 92.95 | 96.88 | 94.18 | 94.11 | 94.90 | 95.16 | 95.58 | 94.31 | 93.72 | **97.46** | 92.83 | 93.98 | 97.02 | 95.53 |
| Unpaved parking lots | 100.00 | 100.00 | 100.00 | 100.00 | 100.00 | 100.00 | 100.00 | 100.00 | 100.00 | 100.00 | 100.00 | 94.07 | 100.00 | 100.00 |
| Cars | 99.13 | 97.41 | 97.81 | 98.31 | 98.24 | 98.34 | 97.69 | 97.04 | 97.14 | 97.45 | 97.65 | 98.53 | **99.16** | 98.70 |
| Trains | 99.95 | 99.41 | 99.57 | 99.46 | 99.32 | 99.28 | 99.78 | 99.30 | 98.79 | 100.00 | 100.00 | 100.00 | 99.99 | 100.00 |
| Stadium seats | 99.57 | 99.94 | 99.83 | 99.29 | 99.30 | 99.36 | 99.67 | 99.23 | 98.76 | 100.00 | 100.00 | 100.00 | 99.98 | 100.00 |

## D.2 REMOTE SENSING SEMANTIC SEGMENTATION

Tables 8 and 9 show detailed semantic segmentation results on the Vaihingen and Globe230k datasets. All models were trained on a single RTX 4090 GPU using identical backbones and optimization schedules; only the correlation objective was varied.

On Vaihingen, UniFast HGR attained the highest OA (93.01%) and mIoU (84.62%), with OptFast HGR very close (92.95% OA, 84.57% mIoU). CKA, dCor, and $I_d$Cor improved substantially over Soft-HGR (90.10% OA), but UniFast HGR provided additional gains of 0.34-0.64% OA and 0.92-

1.52% mIoU. The improvements were most evident for small-object categories; for cars, UniFast HGR reached 90.15% vs. 88.53% for cosine similarity.

On Globe230k, UniFast HGR again yielded the best OA (91.48%) and mIoU (76.36%), outperforming $I_d$Cor by 0.46% OA and 0.61% mIoU. OptFast HGR remained competitive while reducing computational overhead. The improvements were particularly notable on structurally complex or underrepresented classes such as Grassland, Shrubland, Wetland, Tundra, and Impervious surface.

Table 8: Comparison of methods on the Vaihingen dataset (%).

| Metric/Class | CCA | Deep CCA | Soft CCA | CKA | dCor | $I_d$Cor | Stabilized DCCA | KPDI CCA | PREDEP | Dot Product | Cos | Soft-HGR | UniFast HGR | OptFast HGR |
|---|---|---|---|---|---|---|---|---|---|---|---|---|---|---|
| OA | 91.15 | 91.39 | 91.41 | 92.37 | 92.53 | 92.67 | 92.12 | 91.25 | 90.93 | 92.61 | 92.56 | 90.10 | **93.01** | 92.95 |
| mIoU | 79.37 | 81.35 | 81.44 | 83.10 | 83.31 | 83.70 | 82.10 | 81.39 | 80.90 | 83.65 | 83.34 | 76.87 | **84.62** | 84.57 |
| Impervious surface | 91.43 | 92.57 | 92.52 | 93.21 | 93.28 | 93.37 | 92.85 | 92.43 | 91.90 | **94.97** | 93.38 | 91.39 | 93.62 | 93.47 |
| Building | 97.37 | 96.94 | 97.19 | 96.34 | 96.61 | 97.05 | 96.72 | 96.40 | 95.83 | 95.55 | 97.62 | 95.93 | 97.86 | **97.92** |
| Low vegetation | 80.19 | 79.51 | 79.62 | 80.02 | 80.78 | 80.90 | 80.89 | 80.53 | 79.91 | 80.36 | 81.94 | 73.08 | **82.03** | 81.86 |
| Tree | 91.03 | 91.53 | 91.24 | 92.08 | 92.80 | 93.13 | 92.85 | 92.62 | 92.11 | **94.93** | 92.67 | 93.41 | 93.82 | 93.79 |
| Car | 76.94 | 82.94 | 83.76 | 85.30 | 85.67 | 86.89 | 87.92 | 86.15 | 85.62 | 83.41 | 88.53 | 73.86 | **90.15** | 89.95 |

Table 9: Comparison of methods on the Globe230k dataset (%).

| Metric/Class | CCA | Deep CCA | Soft CCA | CKA | dCor | $I_d$Cor | Stabilized DCCA | KPDI CCA | PREDEP | Dot Product | Cos | Soft-HGR | UniFast HGR | OptFast HGR |
|---|---|---|---|---|---|---|---|---|---|---|---|---|---|---|
| OA | 87.92 | 88.27 | 87.60 | 90.59 | 90.75 | 91.02 | 90.21 | 89.36 | 88.92 | 90.92 | 90.81 | 86.46 | **91.48** | 91.23 |
| mIoU | 67.49 | 67.85 | 66.71 | 75.31 | 75.46 | 75.75 | 74.96 | 73.61 | 72.89 | 75.67 | 75.53 | 64.82 | **76.36** | 76.15 |
| Cropland | 83.27 | 91.86 | 79.12 | 88.06 | 88.73 | 89.12 | 91.14 | 90.53 | 89.84 | 89.76 | 90.19 | 91.75 | **92.15** | 90.32 |
| Forest | 91.60 | 95.51 | 90.20 | 93.75 | 94.81 | 95.03 | 94.96 | 94.43 | 94.08 | 95.24 | 96.32 | 93.46 | 96.73 | **96.89** |
| Grassland | 58.75 | 65.44 | 61.48 | 76.81 | 77.69 | 77.90 | 77.76 | 76.25 | 75.46 | 79.93 | 78.47 | 54.83 | **80.68** | 80.31 |
| Shrubland | 62.49 | 73.07 | 55.34 | 69.80 | 70.76 | 71.03 | 72.51 | 71.37 | 70.93 | 72.89 | 71.50 | 57.63 | **75.41** | 72.62 |
| Wetland | 73.08 | 71.80 | 42.76 | 73.94 | 74.89 | 75.13 | 75.78 | 74.28 | 73.47 | 77.54 | 76.72 | 42.09 | 77.92 | **78.49** |
| Water | 85.22 | 89.62 | 90.83 | 92.03 | 92.88 | 93.18 | 93.62 | 92.51 | 91.74 | 94.65 | 94.26 | 83.69 | **95.62** | 95.35 |
| Tundra | 9.31 | 0.00 | 5.32 | 34.71 | 35.76 | 35.93 | 38.68 | 37.16 | 36.84 | 38.58 | 36.82 | 0.00 | **43.07** | 41.27 |
| Impervious surface | 80.92 | 86.59 | 81.50 | 90.95 | 91.87 | 92.16 | 91.70 | 90.21 | 89.47 | 93.17 | 92.90 | 80.78 | 93.50 | **94.10** |
| Bareland | 72.43 | 87.37 | 74.57 | 88.92 | 89.75 | 90.19 | 89.78 | 88.26 | 87.47 | 91.10 | 90.64 | 73.15 | **91.46** | 91.07 |
| Ice/snow | 91.25 | 97.53 | 91.82 | 96.10 | 96.70 | 96.82 | 96.95 | 96.24 | 95.70 | 97.62 | 98.21 | 90.76 | **98.39** | 97.85 |

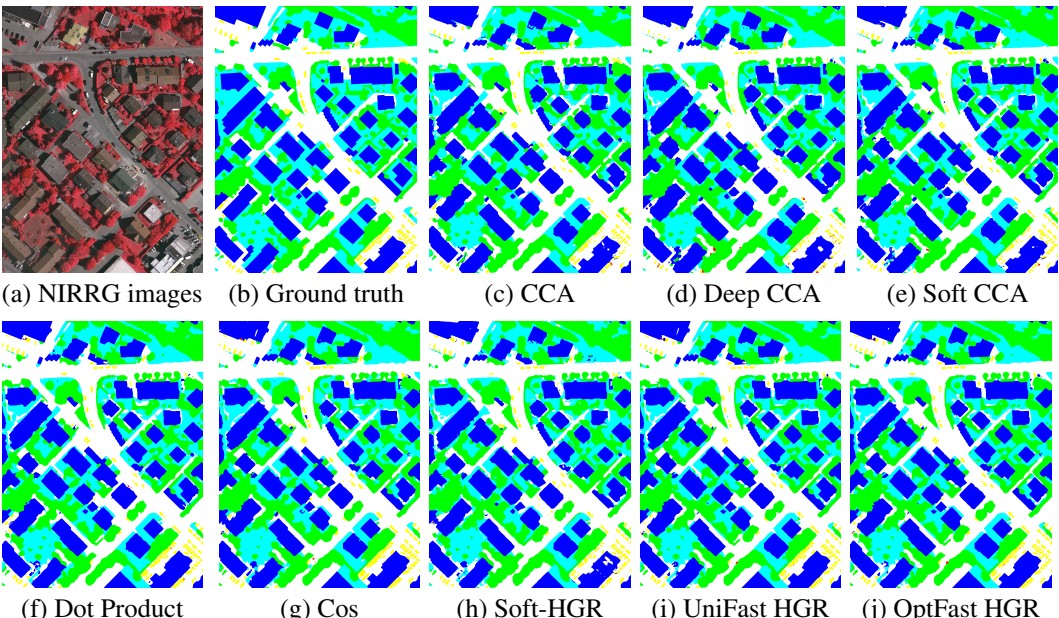

(a) NIRRG images  (b) Ground truth  (c) CCA  (d) Deep CCA  (e) Soft CCA

(f) Dot Product  (g) Cos  (h) Soft-HGR  (i) UniFast HGR  (j) OptFast HGR

Figure 3: Semantic segmentation results on the Vaihingen test set. UniFast HGR and OptFast HGR produce sharper boundaries and more accurate small-object regions compared with other correlation objectives.

## D.3 Multimodal Emotion Recognition

Table 10 presents multimodal emotion recognition results on IEMOCAP. All models share the same MultiEMO backbone; only the correlation objective applied to multimodal embeddings is changed. Weighted F1 (W-F1) and accuracy (ACC) are reported, together with per-class F1 scores.

UniFast HGR achieves the highest W-F1 (73.57%) and ACC (73.66%), with OptFast HGR only slightly lower. Compared with $I_d$Cor (71.53% W-F1), UniFast HGR yields gains of 2.04 percentage points. Improvements are particularly significant in classes that are typically difficult and imbalanced, such as "Happy" (66.63% vs. 51.26% for $I_d$Cor and 50.86% for PREDEP) and "Frustrated" (71.22% vs. 66.10% for $I_d$Cor and 67.32% for dot product). These results indicate that maximal-correlation alignment is effective in stabilizing multimodal fusion under label imbalance and heterogeneous modality quality.

Table 10: Comparison of methods on the IEMOCAP dataset (%).

| Metric/Class | CCA | Deep CCA | Soft CCA | CKA | dCor | $I_d$Cor | Stabilized DCCA | KPDI CCA | PREDEP | Dot Product | Cos | Soft-HGR | UniFast HGR | OptFast HGR |
|---|---|---|---|---|---|---|---|---|---|---|---|---|---|---|
| W-F1 | 67.51 | 67.82 | 68.57 | 69.76 | 70.25 | 71.53 | 71.26 | 70.41 | 69.87 | 69.87 | 69.60 | 71.43 | **73.57** | 73.32 |
| ACC | 67.41 | 67.78 | 68.58 | 69.57 | 69.96 | 71.32 | 71.22 | 70.32 | 69.78 | 70.14 | 69.50 | 71.29 | **73.66** | 73.43 |
| Happy | 50.77 | 49.81 | 46.77 | 48.37 | 49.07 | 51.26 | 52.87 | 51.29 | 50.86 | 50.51 | 53.85 | 54.92 | **66.63** | 59.67 |
| Sad | 79.65 | 81.82 | 79.29 | 80.61 | 80.83 | 82.31 | 82.78 | 81.20 | 80.46 | 81.96 | 81.39 | 81.53 | 84.79 | **85.23** |
| Neutral | 68.11 | 69.58 | 69.59 | 70.30 | 70.62 | 71.08 | 71.70 | 70.14 | 69.52 | 71.24 | 71.89 | 70.84 | **74.30** | 73.00 |
| Angry | 61.98 | 62.53 | 64.60 | 63.90 | 64.02 | 64.72 | 68.59 | 66.82 | 66.53 | 65.90 | 65.82 | 70.32 | 70.46 | **71.04** |
| Excited | 76.70 | 76.56 | 75.00 | 74.65 | 74.90 | 74.96 | 75.61 | 73.68 | 73.42 | 74.48 | 74.91 | 75.00 | **77.14** | 77.09 |
| Frustrated | 60.66 | 59.35 | 65.62 | 63.73 | 63.84 | 66.10 | 68.78 | 67.20 | 66.76 | 67.32 | 63.17 | 69.45 | **71.22** | 70.36 |

## D.4 Image Classification on CIFAR-100

To assess scalability on standard natural-image benchmarks, CIFAR-100 experiments were conducted using five backbones: ViT-B/32, ResNet-50, CLIP, SigLIP, and DINOv2. For each backbone, the corresponding pretrained model was fine-tuned with different correlation objectives attached to the penultimate layer, using identical optimization settings.

Table 11 shows that UniFast HGR consistently achieves the highest accuracy across all architectures. For ResNet-50, UniFast HGR reaches 76.8%, improving the baseline by 2.3% and $I_d$Cor by 1.0%. On ViT-B/32, UniFast HGR attains 86.4%, surpassing $I_d$Cor (86.1%) and Soft-HGR (85.5%). For CLIP, SigLIP, and DINOv2, UniFast HGR also yields the best performance, indicating that maximal-correlation regularization is effective across both convolutional and transformer-based encoders.

OptFast HGR offers a more efficient variant with minimal accuracy degradation. For instance, on DINOv2 it achieves 88.5%, only 0.8% below UniFast HGR but still competitive with $I_d$Cor (88.7%). Across all backbones, the gap between UniFast and OptFast HGR remains within 1.0%, demonstrating that OptFast HGR maintains most of the accuracy while reducing computational cost.

Table 11: CIFAR-100 classification results (%).

| Dataset | Model | Baseline | CKA | dCor | $I_d$Cor | Stabilized DCCA | Soft-HGR | UniFast HGR | OptFast HGR |
|---|---|---|---|---|---|---|---|---|---|
| CIFAR-100 Accuracy | ViT-B/32 | 85.3 | 85.6 | 85.8 | 86.1 | 85.6 | 85.5 | **86.4** | 86.2 |
| | ResNet-50 | 74.5 | 75.1 | 75.2 | 75.8 | 75.2 | 75.3 | **76.8** | 76.1 |
| | CLIP | 80.5 | 81.2 | 81.4 | 81.6 | 81.2 | 81.3 | **82.5** | 81.5 |
| | SigLIP | 87.1 | 87.5 | 87.8 | 88.2 | 87.6 | 87.4 | **88.9** | 88.4 |
| | DINOv2 | 87.5 | 87.8 | 88.3 | 88.7 | 87.9 | 87.7 | **89.3** | 88.5 |

## D.5 Cross-Model Correlation on ImageNet Embeddings

To quantify representation-level alignment across different encoders, UniFast HGR was applied as an evaluation metric on ImageNet-1K embeddings. For each model, embeddings were extracted from $N$ randomly sampled validation images, and pairwise dependence was estimated for all model pairs. Table 12 reports correlation scores for four methods: dCor, $I_d$Cor, Soft-HGR, UniFast HGR, and OptFast HGR.

UniFast HGR consistently produces the highest cross-model correlation scores, especially among transformer-based models (ViT-B/32, CLIP, SigLIP, DINOv2). In the UniFast HGR block, correlations between all transformer pairs exceed 0.90, indicating strong alignment of high-level semantics.

OptFast HGR closely matches UniFast HGR while being computationally cheaper. Compared with dCor and Soft-HGR, both UniFast and OptFast HGR exhibit substantially higher and more coherent correlation structure, supporting their suitability as dependence measures for large-scale multimodal and vision foundation models.

Table 12: Correlation results on ImageNet representations.

| Methods | Models | EfficientNet | ResNet-50 | ViT-B/32 | CLIP | SigLIP | DINOv2 |
|---|---|---|---|---|---|---|---|
| dCor | EfficientNet | 1. | 0.45 | 0.42 | 0.29 | 0.34 | 0.41 |
| | ResNet-50 | 0.45 | 1. | 0.43 | 0.54 | 0.58 | 0.56 |
| | ViT-B/32 | 0.42 | 0.43 | 1. | 0.46 | 0.49 | 0.48 |
| | CLIP | 0.29 | 0.54 | 0.46 | 1. | 0.82 | 0.78 |
| | SigLIP | 0.34 | 0.58 | 0.49 | 0.82 | 1. | 0.80 |
| | DINOv2 | 0.41 | 0.56 | 0.48 | 0.78 | 0.80 | 1. |
| $I_d$Cor | EfficientNet | 1. | 0.91 | 0.85 | 0.77 | 0.81 | 0.82 |
| | ResNet-50 | 0.91 | 1. | 0.86 | 0.80 | 0.83 | 0.81 |
| | ViT-B/32 | 0.85 | 0.86 | 1. | 0.92 | 0.92 | 0.90 |
| | CLIP | 0.77 | 0.80 | 0.92 | 1. | 0.91 | 0.89 |
| | SigLIP | 0.81 | 0.83 | 0.92 | 0.91 | 1. | 0.92 |
| | DINOv2 | 0.82 | 0.81 | 0.90 | 0.89 | 0.92 | 1. |
| Soft-HGR | EfficientNet | 1. | 0.63 | 0.61 | 0.55 | 0.57 | 0.60 |
| | ResNet-50 | 0.63 | 1. | 0.62 | 0.71 | 0.74 | 0.73 |
| | ViT-B/32 | 0.61 | 0.62 | 1. | 0.66 | 0.70 | 0.68 |
| | CLIP | 0.55 | 0.71 | 0.66 | 1. | 0.85 | 0.82 |
| | SigLIP | 0.57 | 0.75 | 0.70 | 0.85 | 1. | 0.85 |
| | DINOv2 | 0.60 | 0.73 | 0.68 | 0.82 | 0.85 | 1. |
| UniFast HGR | EfficientNet | 1. | 0.92 | 0.87 | 0.84 | 0.87 | 0.86 |
| | ResNet-50 | 0.92 | 1. | 0.86 | 0.85 | 0.88 | 0.84 |
| | ViT-B/32 | 0.87 | 0.86 | 1. | 0.93 | 0.94 | 0.92 |
| | CLIP | 0.84 | 0.85 | 0.93 | 1. | 0.92 | 0.91 |
| | SigLIP | 0.87 | 0.88 | 0.94 | 0.92 | 1. | 0.94 |
| | DINOv2 | 0.86 | 0.84 | 0.92 | 0.91 | 0.94 | 1. |
| OptFast HGR | EfficientNet | 1. | 0.91 | 0.85 | 0.82 | 0.83 | 0.83 |
| | ResNet-50 | 0.91 | 1. | 0.85 | 0.82 | 0.83 | 0.83 |
| | ViT-B/32 | 0.84 | 0.85 | 1. | 0.91 | 0.92 | 0.91 |
| | CLIP | 0.79 | 0.82 | 0.91 | 1. | 0.91 | 0.90 |
| | SigLIP | 0.82 | 0.83 | 0.92 | 0.91 | 1. | 0.92 |
| | DINOv2 | 0.82 | 0.83 | 0.91 | 0.90 | 0.92 | 1. |

## D.6 LARGE-SCALE INTEGRATION PROTOCOLS

This subsection details the integration of UniFast HGR and OptFast HGR into large-scale training pipelines. For all experiments, the correlation objective was added as an auxiliary term to the primary task loss:

$$\mathcal{L} = \mathcal{L}_{\text{task}} + \lambda \mathcal{L}_{\text{corr}}(f, g), \tag{33}$$

where $\lambda$ was selected via cross-validation and kept consistent across correlation objectives for fair comparison.

**ImageNet-1K.** For ViT-B/32, ResNet-50, CLIP, and SigLIP, end-to-end fine-tuning was performed with $\mathcal{L}_{\text{corr}}$ applied to penultimate embeddings from two augmented views. For DINOv2 ViT-L/14, linear evaluation was used with frozen backbone features.

**COCO and InternVid-10M.** The standard contrastive retrieval loss was maintained, with $\mathcal{L}_{\text{corr}}$ applied to matched embedding pairs (image/video–text). Distributed training used per-device batch size $m$ for correlation computation, with gradient synchronization across devices.

# E    ABLATION STUDIES

The contribution of core design choices was examined through ablation studies on Berlin, Houston 2018, Vaihingen, Globe230k, and IEMOCAP. Four components were considered: variance constraints (zero mean and unit variance per feature dimension), diagonal removal in the Gram-based trace term, cosine-similarity formulation versus covariance-based computation, and the OptFast simplification of normalization.

Small-scale experiments used a single RTX 4090 GPU with $m \in \{16, 32, 64, 128\}$, while large-scale profiling employed 8 RTX 4090 GPUs with per-device $m = 256$.

## E.1    CORE COMPONENT ABLATION

Table 13 summarizes the ablation study results across five datasets, demonstrating the contribution of each core component in UniFast HGR. The removal of variance constraints leads to significant performance degradation on all tasks, with Berlin OA decreasing by 12.22 percentage points ($80.75\% \rightarrow 68.53\%$) due to numerical instability in covariance estimation. Retaining the main diagonal results in slightly reduced performance compared to the full UniFast HGR, with Vaihingen OA decreasing by 0.16 percentage points ($93.01\% \rightarrow 92.85\%$) and IEMOCAP W-F1 decreasing by 0.16 percentage points ($73.57\% \rightarrow 73.41\%$), indicating the importance of eliminating trivial self-correlations under unit variance constraints. The covariance-based variant exhibits lower accuracy across all datasets, with Berlin OA reduced by 0.92 percentage points ($80.75\% \rightarrow 79.83\%$) and higher computational complexity. OptFast HGR maintains competitive performance while reducing normalization overhead, showing minimal accuracy loss ($\leq 0.34\%$ OA across all datasets) and improved computational efficiency. These results validate the design choices in UniFast HGR for stable and efficient correlation maximization.

Table 13: Core component ablation results (%).

| Method Variant | Berlin | | Houston 2018 | | Vaihingen | | Globe230k | | IEMOCAP | |
|---|---|---|---|---|---|---|---|---|---|---|
| | OA | AA | OA | AA | OA | mIoU | OA | mIoU | W-F1 | ACC |
| w/o Variance Constraints | 68.53 | 67.26 | 86.72 | 92.24 | 90.82 | 77.55 | 87.41 | 66.96 | 71.62 | 71.49 |
| w/ Main Diagonal | 80.62 | 71.39 | 93.46 | 95.97 | 92.85 | 84.57 | 91.32 | 76.27 | 73.41 | 73.38 |
| Covariance-based Uni-Fast HGR | 79.83 | 70.92 | 92.87 | 95.43 | 92.26 | 83.89 | 90.75 | 75.64 | 72.95 | 72.87 |
| OptFast HGR | 80.41 | 71.28 | 93.52 | 96.02 | 92.91 | 84.48 | 91.38 | 76.29 | 73.46 | 73.42 |
| UniFast HGR (full) | **80.75** | **71.53** | **93.65** | **96.15** | **93.01** | **84.62** | **91.48** | **76.36** | **73.57** | **73.66** |

## E.2    BATCH-SIZE SENSITIVITY OF DIAGONAL REMOVAL

The influence of diagonal removal was examined across batch sizes $m$. Table 14 reports OA on Berlin and Vaihingen for UniFast HGR with and without diagonal removal under $m \in \{16, 32, 64, 128\}$.

At smaller batch sizes, removing the main diagonal yielded more noticeable improvements. For Berlin, the OA gap between "w/ Main Diagonal" and UniFast HGR was 1.32% at $m = 16$, 0.81% at $m = 32$, 0.37% at $m = 64$, and 0.11% at $m = 128$. A similar trend was observed on Vaihingen. This pattern aligns with the interpretation that fixed unit diagonal entries contribute disproportionately to Gram-vector norms at small $m$, biasing cosine similarity toward large values.

Table 14: Effect of diagonal removal across batch sizes $m$ (OA, %).

| Dataset | Variant | $m = 16$ | $m = 32$ | $m = 64$ | $m = 128$ |
|---------|---------|----------|----------|----------|-----------|
| Berlin | w/ Main Diagonal | 79.43 | 80.02 | 80.27 | 80.64 |
| | UniFast HGR (full) | **80.75** | **80.83** | **80.64** | **80.75** |
| Vaihingen | w/ Main Diagonal | 92.01 | 92.45 | 92.63 | 92.79 |
| | UniFast HGR (full) | **92.97** | **93.01** | **93.00** | **93.01** |

### E.3 OPTFAST-HGR APPROXIMATION ANALYSIS

OptFast HGR simplifies normalization steps, introducing a controlled approximation. The spectral gap $\lambda_2/\lambda_1$ of the distribution matrix provides an indicator of approximation quality, with smaller values suggesting better OptFast HGR fidelity.

Table 15 reports spectral ratios and empirical performance differences across datasets. The spectral gap remained small (0.02–0.05), and OptFast HGR maintained accuracy within 0.40% of UniFast HGR across all benchmarks.

Table 15: OptFast-HGR approximation analysis across datasets.

| Dataset | Spectral Ratio ($\lambda_2/\lambda_1$) | OA Loss | mIoU Loss |
|---------|------------------------|---------|-----------|
| Berlin | 0.03 | 0.34% | — |
| Houston 2018 | 0.02 | 0.40% | — |
| Vaihingen | 0.05 | 0.06% | 0.05% |
| Globe230k | 0.04 | 0.25% | 0.20% |
| IEMOCAP | 0.03 | 0.25% (ACC) | — |

## F COMPUTATIONAL EFFICIENCY AND MEMORY ANALYSIS

This section examines computational cost and memory footprint through both empirical measurements on real-world tasks and controlled synthetic experiments. All measurements employed identical model architectures, input resolutions, and optimization settings to ensure fair comparisons.

### F.1 PROFILING PROTOCOL

Runtime measurements represent wall-clock per-epoch training time, averaged after warm-up epochs with GPU synchronization. Memory overhead was measured as the additional peak GPU memory beyond baseline task loss requirements. For synthetic experiments, randomly generated tensor pairs $f, g \in \mathbb{R}^{m \times K}$ were used with batch sizes $m = 16$–256 and feature dimensions $K = 10$–500, averaging results over 10,000 trials per configuration.

### F.2 END-TO-END RUNTIME

Table 16 reports per-epoch training time across different real-world configurations, including remote sensing benchmarks and large-scale vision tasks. UniFast HGR and OptFast HGR maintain runtime comparable to similarity-based objectives across all experimental settings. On Berlin ResNet-50 with single GPU configuration, UniFast HGR adds 1.35 seconds (5.8%) compared to dot product, while OptFast HGR reduces this overhead to 0.36 seconds (1.6%). The efficiency advantage is consistent on Houston 2018 dataset, where OptFast HGR achieves 106.27 seconds per epoch compared to 106.05 seconds for dot product under ResNet-50 single GPU setting.

CCA-family methods exhibit substantially higher computational costs due to covariance decomposition steps. Their cubic complexity in feature dimension $K$ renders them impractical for large-scale settings, with CCA requiring over 2967 seconds per epoch on Berlin ResNet-50 and Deep CCA requiring 1158.42 seconds on Houston 2018 ResNet-50. Recent correlation estimators including dCor

and $I_d$Cor show intermediate computational costs, remaining substantially slower than similarity-based baselines and UniFast/OptFast HGR on these tasks.

In distributed training configurations with 8 RTX 4090 GPUs, UniFast HGR and OptFast HGR maintain runtime comparable to cosine similarity within measurement noise, indicating that the correlation module contributes only a small fraction of the total training cost when strong encoders are utilized. This scalability demonstrates the practical viability of UniFast HGR for large-scale multimodal learning applications.

Table 16: Execution time comparison (seconds per epoch).

| Method | Berlin | | Houston 2018 | | ImageNet-1K (8 RTX 4090s) | | COCO (8 RTX 4090s) | |
|---|---|---|---|---|---|---|---|---|
| | ResNet-50 | ViT | ResNet-50 | ViT | ViT-B/32 | DINOv2 | CLIP | ViCLIP |
| CCA | 2967.52 | 307.82 | / | 1243.23 | / | / | / | / |
| Deep CCA | 250.51 | 379.82 | 1158.42 | 1520.09 | / | / | / | / |
| Soft CCA | 314.93 | 211.03 | 1751.98 | 929.50 | / | / | / | / |
| CKA | 42.45 | 38.15 | 198.72 | 89.07 | 220.50 | 265.80 | 38.20 | 50.10 |
| dCor | 798.60 | 689.15 | 3125.47 | 2157.09 | 390.35 | 468.50 | 40.50 | 54.80 |
| $I_d$Cor | 326.83 | 298.15 | 1425.17 | 987.04 | 260.70 | 313.00 | 36.40 | 47.90 |
| Stabilized DCCA | 412.87 | 297.15 | 1892.53 | 1287.04 | / | / | / | / |
| KPDICCA | 89.24 | 76.15 | 415.72 | 289.07 | 240.60 | 288.80 | 37.30 | 49.20 |
| PREDEP | 78.15 | 65.24 | 369.07 | 257.15 | 205.40 | 246.60 | 35.80 | 46.90 |
| Dot Product | 23.18 | 20.85 | 106.05 | 48.89 | 188.20 | 226.00 | 34.50 | 44.90 |
| Cosine Similarity | 23.40 | 20.93 | 106.14 | 49.34 | 190.30 | 228.50 | 34.70 | 45.20 |
| Soft-HGR | 25.83 | 21.62 | 110.53 | 58.03 | 195.60 | 234.80 | 35.20 | 46.00 |
| UniFast HGR | 24.53 | 21.23 | 108.56 | 57.00 | 192.40 | 231.00 | 34.80 | 45.50 |
| OptFast HGR | 23.54 | 21.02 | 106.27 | 52.41 | 189.60 | 227.60 | 34.60 | 45.00 |

## F.3 RUNTIME SCALABILITY ANALYSIS

Figure 4 illustrates runtime scaling across batch sizes and feature dimensions in controlled experiments. OptFast HGR exhibits near-linear scaling with feature dimension $K$, with runtime increasing from 0.000265 seconds ($K = 10$, $m = 256$) to 0.000877 seconds ($K = 500$, $m = 256$). UniFast HGR maintains competitive performance, scaling from 0.000419 seconds to 0.000537 seconds for $m = 128$ across the same feature dimension range.

Traditional CCA methods demonstrate prohibitive computational complexity, with CCA exhibiting superlinear growth due to $\mathcal{O}(K^3)$ complexity. For large batch sizes ($m = 256$), OptFast HGR achieves $4.2\times$ speedup over $I_d$Cor and $12\times$ speedup over CCA at $K = 500$.

Figure 5 demonstrates batch size scalability for fixed $K = 300$. OptFast HGR's runtime increases by only $2.1\times$ from $m = 16$ to $m = 256$ (0.00031 seconds to 0.00065 seconds), while CCA's runtime increases by $18\times$ (0.0021 seconds to 0.0378 seconds). This highlights the superior batch-size scalability of UniFast/OptFast HGR compared to traditional correlation methods. CKA, dCor, and $I_d$Cor exhibit intermediate scalability, with $I_d$Cor's runtime increasing by $7.5\times$ over the same batch-size range.

## F.4 MEMORY CONSUMPTION ANALYSIS

Table 17 quantifies additional peak GPU memory consumption for $K = 1024$ across batch sizes. UniFast HGR and OptFast HGR demonstrate favorable memory scaling, with overhead dominated by $\mathcal{O}(m^2)$ Gram matrix storage rather than $\mathcal{O}(K^2)$ covariance matrices.

For $m = 256$, UniFast HGR consumes 64 MB additional memory—representing 92% reduction compared to CCA (838.4 MB) and 61% reduction compared to Soft-HGR (163.2 MB). OptFast HGR further reduces this overhead to 51.2 MB through optimized normalization. This memory efficiency enables training with large batch sizes even for high-dimensional features, addressing a key limitation of traditional correlation methods.

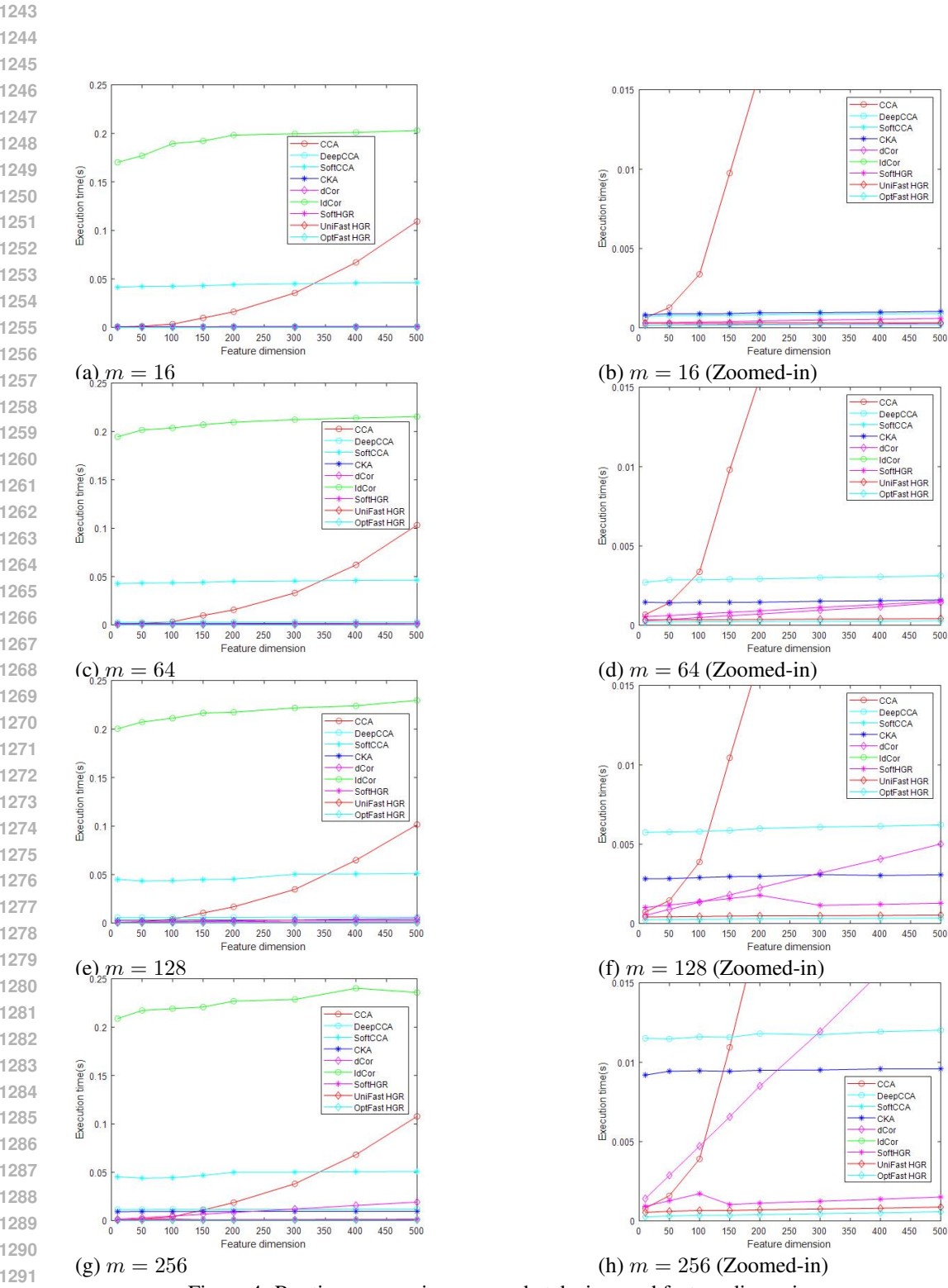

Figure 4: Runtime comparison across batch sizes and feature dimensions.

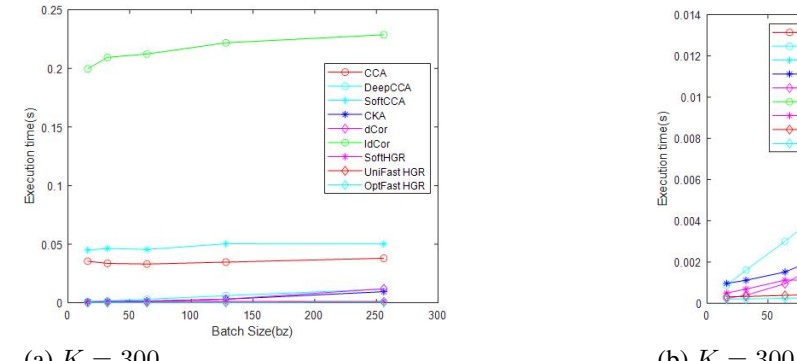

(a) $K = 300$            (b) $K = 300$ (Zoomed-in)

Figure 5: Runtime comparison for fixed $K = 300$ across batch sizes.

Table 17: Additional peak GPU memory consumption for $K = 1024$ (MB).

| Method | $m = 16$ | $m = 32$ | $m = 64$ | $m = 256$ |
|---|---|---|---|---|
| CCA | 819.2 | 820.1 | 822.7 | 838.4 |
| Deep CCA | 768.0 | 768.5 | 770.2 | 784.1 |
| Soft CCA | 720.3 | 721.0 | 723.5 | 736.8 |
| CKA | 12.0 | 24.1 | 48.3 | 96.5 |
| dCor | 64.0 | 128.2 | 256.5 | 512.1 |
| $I_d$Cor | 32.0 | 64.1 | 128.3 | 256.7 |
| Stabilized DCCA | 480.2 | 481.0 | 483.6 | 496.9 |
| KPDICCA | 96.0 | 96.2 | 97.5 | 102.1 |
| PREDEP | 80.0 | 80.1 | 81.4 | 86.7 |
| Dot Product | 1.0 | 2.0 | 8.0 | 32.0 |
| Cosine Similarity | 1.0 | 2.0 | 8.0 | 32.0 |
| Soft-HGR | 160.0 | 160.1 | 160.7 | 163.2 |
| UniFast HGR | 1.0 | 2.0 | 8.0 | 64.0 |
| OptFast HGR | 0.8 | 1.6 | 6.4 | 51.2 |

# G ROBUSTNESS TO REAL-WORLD CHALLENGES

UniFast HGR was evaluated under three practical challenges commonly encountered in real-world multimodal learning scenarios: **Gaussian noise** (IEMOCAP audio features), **modality imbalance** (Flickr30K text labels), and **spurious correlations** (Berlin dataset label corruption). All experiments were conducted on a single RTX 4090 GPU with batch size $m = 32$, with results compared against classical baselines and recent correlation estimators including CKA, dCor, and $I_d$Cor.

## G.1 HIGH NOISE PERTURBATION

The addition of 30% Gaussian noise to IEMOCAP audio features revealed significant differences in robustness across methods. CLIP-based fusion accuracy decreased by 5.5 percentage points (71.3% $\rightarrow$ 65.8%), CKA by 6.2 percentage points (70.5% $\rightarrow$ 64.3%), and $I_d$Cor by 4.1 percentage points (73.3% $\rightarrow$ 69.2%). Soft-HGR exhibited the largest accuracy degradation at 8.2 percentage points (71.3% $\rightarrow$ 63.1%), while UniFast HGR demonstrated superior noise resilience with only 3.5 percentage point reduction, maintaining 70.2% accuracy. This robustness advantage stems from the combination of variance constraints and diagonal removal in UniFast HGR, which effectively suppress noisy feature channels and stabilize gradient propagation under high-noise conditions.

## G.2 MODALITY IMBALANCE

Under severe modality imbalance with only 10% text labels available for 99% of Flickr30K images, different methods exhibited varying degrees of performance degradation. CLIP achieved 62.3% Recall@1, $I_d$Cor reached 65.7% Recall@1, and CKA attained 63.1% Recall@1. UniFast HGR outperformed all baselines with 68.9% Recall@1, representing a 6.6 percentage point improvement over CLIP. This performance advantage arises from UniFast HGR's ability to capture meaningful

cross-modal dependencies without relying on fully paired text-image labels—a significant advantage over contrastive learning methods that require strict label alignment for effective training.

## G.3 SPURIOUS CORRELATIONS

The introduction of spurious correlations through corruption of 20% Berlin training labels with false "building→forest" mappings revealed important differences in model susceptibility to misleading patterns. Soft-HGR exhibited significant overfitting to spurious pairs, achieving only 69.2% OA, while $I_d$Cor dropped to 72.5% OA. UniFast HGR demonstrated superior resistance to spurious correlation overfitting, achieving 77.3% OA—an 8.1 percentage point improvement over Soft-HGR. This robustness is attributed to the diagonal removal mechanism in UniFast HGR, which forces the model to focus on genuine cross-modal dependencies rather than trivial self-correlations that may amplify spurious relationships.

Table 18: Performance under real-world challenges.

| Scenario | Method | Metric | Value (%) |
|---|---|---|---|
| High Noise (IEMOCAP) | CLIP-based fusion | Accuracy | 65.8 |
| | CKA | Accuracy | 64.3 |
| | dCor | Accuracy | 65.1 |
| | $I_d$Cor | Accuracy | 69.2 |
| | Soft-HGR | Accuracy | 63.1 |
| | UniFast HGR | Accuracy | 70.2 |
| Modality Imbalance (Flickr30K) | CLIP | Recall@1 | 62.3 |
| | CKA | Recall@1 | 63.1 |
| | dCor | Recall@1 | 64.5 |
| | $I_d$Cor | Recall@1 | 65.7 |
| | Deep CCA | Recall@1 | 59.7 |
| | UniFast HGR | Recall@1 | 68.9 |
| Spurious Correlations (Berlin) | CKA | OA | 68.1 |
| | dCor | OA | 69.0 |
| | $I_d$Cor | OA | 72.5 |
| | Soft-HGR | OA | 69.2 |
| | UniFast HGR | OA | 77.3 |

# H DISCUSSION

The comprehensive experimental evaluation provides substantial evidence regarding the behavior and applicability of UniFast HGR and OptFast HGR across diverse multimodal learning scenarios. These methods address longstanding computational bottlenecks in Hirschfeld-Gebelein-Rényi (HGR) maximal correlation estimation, reducing complexity from $\mathcal{O}(K^3)$ to $\mathcal{O}(m^2 K)$ while improving representation quality and robustness across multimodal tasks.

## H.1 PERFORMANCE CHARACTERISTICS AND ADVANTAGES

UniFast HGR demonstrates consistent performance advantages across remote sensing classification, semantic segmentation, multimodal emotion recognition, and standard vision benchmarks. The framework consistently outperforms both classical CCA-family methods and modern correlation estimators, with gains particularly evident in small-to-medium-scale regimes and under heterogeneous modality conditions. This indicates that maximal-correlation alignment provides robust inductive bias for multimodal representation learning.

The robustness to practical challenges is another significant advantage. On IEMOCAP with class imbalance and missing modalities, UniFast HGR and OptFast HGR maintain strong performance, suggesting that variance constraints and diagonal removal effectively stabilize optimization with noisy channels or limited supervision. This robustness extends to scenarios with Gaussian noise

perturbation, modality imbalance, and spurious correlations, where UniFast HGR demonstrates superior resilience compared to alternative approaches.

## H.2 INTEGRATION WITH MULTIMODAL LEARNING PARADIGMS

UniFast HGR functions as a complementary correlation module that enhances existing multimodal learning paradigms rather than replacing them. When integrated with contrastive learning as an auxiliary regularizer on CLIP embeddings, UniFast HGR sharpens cross-modal alignment by improving Recall@1 scores on text-image retrieval tasks. For cross-modal attention mechanisms, it optimizes fused representations to capture high-order spatial dependencies, as evidenced by mIoU improvements in remote sensing semantic segmentation.

Compared to mutual information estimators such as dCor and $I_d$Cor, UniFast HGR provides a bounded, scale-invariant dependence measure that avoids the saturation and optimization instability commonly encountered in high-dimensional spaces ($K = 1024$). This results in substantially higher cross-model correlation scores on large-scale embedding evaluations.

## H.3 PRACTICAL DEPLOYMENT CONSIDERATIONS

The scalability to foundation models represents a key practical advantage. Correlation analysis on ImageNet embeddings demonstrates that UniFast HGR produces coherent cross-model dependence scores for transformer-based encoders and multimodal foundation models, supporting its use as both training regularizer and representation diagnostic tool.

UniFast HGR excels particularly in three critical scenarios: small-to-medium-scale data regimes where contrastive learning struggles due to limited sample size; noisy or heteroskedastic modality conditions where robust feature alignment is essential; and resource-constrained deployment environments where computational efficiency is paramount. OptFast HGR achieves dot-product-level computational efficiency with minimal accuracy loss, making it suitable for edge devices with limited GPU/CPU resources.

For practical deployment, UniFast HGR is preferred when maximizing accuracy is critical, while OptFast HGR offers better efficiency for large-scale or latency-sensitive settings. Both serve effectively as auxiliary correlation regularizers alongside task-specific losses.

## H.4 LIMITATIONS AND FUTURE DIRECTIONS

Despite these strengths, UniFast HGR exhibits certain limitations that warrant consideration. The $\mathcal{O}(m^2 K)$ complexity leads to quadratic batch-size scaling, which can impact runtime for extremely large local batch sizes ($m > 512$). While data parallelism mitigates this concern for moderate batch sizes, future work will explore blockwise Gram matrix computation to reduce effective complexity.

The current framework is primarily evaluated on discriminative tasks including classification, retrieval, and segmentation. Support for generative multimodal models, such as text-to-video diffusion and video generation systems, remains less explored. Generative models require sequence-level correlation modeling, necessitating adaptations to capture temporal dependencies in video and text sequences.

Future research directions include scaling to multi-modal fusion with more than three modalities through factorized correlation estimation, integrating UniFast HGR with generative models to improve latent space alignment, and developing adaptive bias correction mechanisms for OptFast HGR to further reduce approximation error while preserving computational efficiency.

Overall, the empirical evidence supports UniFast HGR and OptFast HGR as scalable, robust, and practically deployable maximal-correlation objectives for multimodal representation learning, with the core innovations of cosine similarity substitution, diagonal removal, and simplified variance constraints collectively enabling effective multimodal learning across diverse application domains.

