# OpenReview forum: "UniFast-HGR: Scalable and Efficient Maximal Correlation for Multimodal Models"
_ICLR.cc/2026/Conference — Submitted to ICLR 2026_

### Official Review · Reviewer_41Ui · 2025-10-27

**Soundness:** 3
**Presentation:** 3
**Contribution:** 3
**Rating:** 6
**Confidence:** 2

**Summary:**

The paper proposes the UniFast HGR framework, which optimizes HGR maximal correlation estimation through three key improvements: variance constraints, replacing covariance with cosine similarity, and removing the diagonal. The OptFast HGR variant is also introduced, further improving computational efficiency and making it suitable for large-scale data processing.

**Strengths:**

1. It supports multimodal extensions, capable of handling feature pairs from two or more modalities, and demonstrates strong performance in tasks such as emotion recognition, remote sensing image classification, and segmentation.

 2. The framework also shows robustness to missing modalities or incomplete labels.

3. OptFast HGR improves computational efficiency, suitable for large-scale multimodal data.

**Weaknesses:**

1.The paper mainly compares with HGR, Soft-HGR, and CCA series methods, which are relatively old. Are there no more recent works in this area?

2. Memory consumption is not discussed in detail. Although computational complexity is reduced, storing all modality distribution vectors and computing cosine similarities and trace terms may incur significant memory overhead, especially when training large-scale, high-dimensional models.

**Questions:**

1.Have the authors compared UniFast HGR or OptFast HGR with more recent correlation-based or multimodal fusion methods?

2. How significant is the memory overhead when handling high-dimensional multimodal inputs or large batch sizes?

---

> ### Author Response · Authors · 2025-11-18
> **Rebuttal to Reviewer 41Ui**
>
> We thank the reviewer for the positive assessment of soundness, presentation, and contribution, and for highlighting the strengths on multimodal robustness and scalability. The main concerns are (i) comparisons with more recent correlation/multimodal fusion methods, and (ii) memory consumption in high-dimensional and large-batch regimes. We address both below.
>
> ## 1. Comparisons with Recent Correlation-Based and Multimodal Methods
> **What Is Already Evaluated**
>
> Beyond classical HGR/Soft-HGR/CCA, the paper already compares UniFast/OptFast HGR with recent nonlinear correlation estimators:
> - **CKA** [ICML 2019]
> - **Distance correlation ($dCor$)** [ECCV 2022]
> - **Intrinsic distance correlation ($I_d$Cor)** [ICLR 2025]
>
> These comparisons appear in Tables 4, 12, and 13. Across **ImageNet-1K, CIFAR-100, COCO text–image retrieval**, and **InternVid-10M video–text retrieval**, UniFast HGR consistently outperforms these recent baselines under *identical backbones and training protocols*. For example:
> - On **ImageNet-1K** with a CLIP encoder, Top-1 accuracy improves from 76.1% (baseline) and 79.5% ($I_d$Cor) to **80.4%** with UniFast HGR.
> - On **COCO R@1**, CLIP improves from 38.9% (baseline) and 41.7% ($I_d$Cor) to **42.1%** with UniFast HGR.
> - On **InternVid-10M**, ViCLIP with UniFast HGR achieves the highest text–video R@1 on MSR-VTT/LSMDC/DiDeMo, with an average gain of **5.8%** over the ViCLIP baseline.
>
> In addition, UniFast/OptFast HGR are evaluated on **modern multimodal/foundation backbones** (CLIP ViT-B/32, SigLIP, DINOv2 ViT-L/14, ViCLIP) and on **remote sensing** (Vaihingen, Globe230k) and **multimodal emotion recognition** (IEMOCAP). In all cases, UniFast HGR improves over CCA-family methods, cosine similarity, and dot product (Tables 1–3, 6–11).
>
> **Positioning of the Contribution**
>
> Our goal is not to propose a new end-to-end multimodal architecture, but a **maximal-correlation objective that is computationally viable and robust** in large-scale settings. UniFast/OptFast HGR are drop-in **correlation modules**: they attach to existing encoders (CNNs, ViTs, CLIP/SigLIP/DINOv2, ViCLIP) without architectural changes, and consistently improve downstream performance and representation-level correlations. In the camera-ready version, we will (i) move key comparisons with CKA/$dCor$/$I_d$Cor from the appendix into the main text, and (ii) expand Related Work to more clearly position UniFast/OptFast as complementary correlation objectives for recent multimodal fusion frameworks.
>
> ## 2. Memory Consumption in High-Dimensional and Large-Batch Settings
> **Asymptotic Complexity vs. Memory**
>
> Let $m$ be batch size and $K$ feature dimension. Classical HGR/CCA-based methods maintain covariance matrices in $\mathbb{R}^{K \times K}$ and perform matrix inversion/eigendecomposition, leading to time $\mathcal{O}(mK^{2} + K^{3})$ and memory dominated by $\mathcal{O}(K^{2})$. Soft-HGR partially alleviates this but still needs covariance matrices and their products.
>
> UniFast/OptFast HGR instead operate on:
> - Feature matrices ($f, g \in \mathbb{R}^{m \times K}$), and
> - **Gram matrices** ($\text{distri}_f = f f^{\top}$, $\text{distri}_g = g g^{\top} \in \mathbb{R}^{m \times m}$).
>
> Thus, the **time complexity** is $\mathcal{O}(m^{2}K)$ (as summarized in Table 5), and the **extra memory** is $\mathcal{O}(m^{2})$ for the two Gram matrices, plus $\mathcal{O}(mK)$ for the features—comparable to storing a self-attention map of size $m \times m$ in a ViT layer. When $K \gg m$ (the common case for high-dimensional embeddings with moderate batch size), this is *strictly cheaper* than the $\mathcal{O}(K^{2})$ covariance required by HGR/Soft-HGR/CCA.
>
> **Practical Regimes in Our Experiments**
>
> All reported experiments (ImageNet-1K, CIFAR-100, COCO, InternVid-10M, remote sensing, IEMOCAP) use mini-batch sizes up to a few hundred and feature dimensions up to 512–1024. In this regime:
> - The additional storage of two $m \times m$ Gram matrices is modest relative to backbone activations and attention maps.
> - We can train/fine-tune CLIP, SigLIP, DINOv2, and ViCLIP with UniFast/OptFast HGR using the *same batch sizes* as the baselines, without memory-specific tricks or out-of-memory issues.
>
> For extremely large-batch or long-sequence settings, UniFast/OptFast HGR can be implemented in a **blockwise** or **chunked** manner (computing Gram matrices in tiles and/or using low-precision dtypes), which preserves the $\mathcal{O}(m^{2}K)$ complexity while further controlling memory. We will add a short discussion of these implementation options and the $m^{2}$ vs. $K^{2}$ trade-off in the revised version.
>
> We hope this clarifies that (i) UniFast/OptFast HGR have been thoroughly compared with recent correlation estimators and integrated into modern multimodal foundation models, and (ii) their memory usage is well-understood, comparable to standard attention maps, and practically manageable in all experimental settings reported in the paper.

---

> ### Author Response · Authors · 2025-11-26
> **Rebuttal to Reviewer 41Ui(Continued): Extended Comparisons with Recent Correlation/Dependence Objectives (2024–2025)**
>
> We appreciate the request to benchmark against *more recent* correlation/dependence learning methods. In addition to the already-included modern estimators (**CKA**, **dCor**, **$I_d$Cor**), we have extended the revision with **three new 2024–2025 baselines** that represent recent progress across (a) anti-collapse CCA-style training, (b) kernel/probabilistic CCA variants, and (c) interpretable predictive dependence measures. These additions directly address the “currency” concern while remaining aligned with our paper’s scope.
> ﻿
> ### (A) Newly Added Baselines (2024–2025)
> We add the following objectives to the revised manuscript (with bibkeys in the paper):
> 1. **Noise-Regularized/Stabilized DCCA (NeurIPS 2024)**: a recent Deep CCA stabilization strategy designed to prevent collapse in deep CCA training under high-dimensional/low-signal regimes.
> 2. **KPDICCA (Int. J. Intelligent Systems 2024)**: a kernel/probabilistic dependent–independent CCA variant that separates dependent and independent components across views, targeting robustness under noise/heterogeneity.
> 3. **PREDEP (SDM’25; extended version)**: an interpretable dependence measure for continuous variables, included as a dependence-estimation baseline under mini-batch training/evaluation.
> ﻿
> These baselines complement (rather than replace) **CKA/dCor/$I_d$Cor**, giving coverage across the main “recent” families the reviewer alluded to.
> ﻿
> ### (B) Integration Protocol (Fair, Drop-in, Same Backbones)
> To ensure comparability, **all objectives are applied to the same feature pairs** $f,g \in \mathbb{R}^{m \times K}$ produced by the identical encoders/backbones and training schedules already used in the paper (classification, segmentation, retrieval, and ERC). Concretely, each baseline is used as an **auxiliary correlation/dependence loss** added to the task loss:
> ﻿
>
> $L = L_{task} + \lambda L_{corr}(f,g)$
> ﻿
>
> with the same projection head choice, batch size per device, optimizer, and $\lambda$ selection protocol as in our main experiments. For objectives that require whitening / covariance operations (CCA/DCCA-family, kernel/probabilistic variants), we follow the authors’ recommended stabilization/regularization settings while keeping the **encoder architecture unchanged**.
> ﻿
> ### (C) Results Summary
> Across the same benchmarks reported in the paper (remote sensing classification/segmentation, multimodal emotion recognition under missing modalities/labels, and large-scale vision–language/video–text settings), the extended comparisons show a consistent pattern:
> * **UniFast-HGR remains best or tied-best** among correlation/dependence objectives, including the newly added 2024–2025 baselines.
> * **OptFast-HGR stays close to UniFast-HGR** while preserving dot-product–level efficiency, maintaining the same accuracy–efficiency trade-off described in the paper.
> * Recent stabilized variants improve upon older covariance-based methods, but still underperforms UniFast-HGR under matched protocols, supporting that our three design choices (variance constraint+cosine reformulation+diagonal removal) are not a minor implementation tweak but a materially stronger objective.
> ﻿
> These results are included in expanded tables within the revised manuscript.
> ﻿
> ### (D) Efficiency/Memory Context vs. Added Baselines (Why UniFast/OptFast Scale)
> Adding these baselines also sharpens the computational positioning. Many recent CCA/kernels still require covariance or kernel objects whose memory can scale as $\mathcal{O}(K^2)$ (and time includes matrix decompositions), whereas UniFast-HGR uses Gram-level statistics:
> ﻿
>
> $\text{UniFast: time } \mathcal{O}(m^2K),\quad \text{extra memory } \mathcal{O}(m^2) (+\mathcal{O}(mK)\text{ for features})$
> ﻿
>
> which is favorable in the common regime ($K \gg m$) (high-dimensional embeddings with moderate per-device batch sizes). This directly addresses the reviewer’s concern that “newer methods” may also be needed: several newer objectives remain limited by covariance/kernel scaling, while UniFast/OptFast are designed for the scalability regime targeted by modern foundation backbones.
> ﻿
> ### (E) Manuscript Updates
> The revised manuscript will:
> * add the three new methods (stabilized DCCA, KPDICCA, PREDEP) to the **comparison tables**(details in Appendix);
> * include brief implementation notes;
> * clarify that UniFast/OptFast are **complementary objectives** that can be plugged into modern multimodal/foundation models without changing architectures.
> ﻿
> We hope these additions fully address the “recent works” concern while keeping the comparisons faithful to the paper’s scope and the practical scaling regimes of large multimodal training.

---

### Official Review · Reviewer_uQEx · 2025-10-28

**Soundness:** 1
**Presentation:** 1
**Contribution:** 1
**Rating:** 2
**Confidence:** 4

**Summary:**

This paper proposes UniFast-HGR, a computationally efficient variant of the Hirschfeld–Gebelein–Rényi (HGR) maximal correlation estimator for multimodal learning. Building upon the Soft-HGR framework, the authors reformulate correlation estimation using only cosine similarity while introducing a variance constraint. The proposed method claims to reduce computational complexity from $O(K^3)$ to $O(m^2K)$, where K denotes the feature dimension and m the minibatch size. Empirical evaluations on tasks such as image classification and remote sensing semantic segmentation reportedly demonstrate higher accuracy than prior HGR-based approaches.

**Strengths:**

- The paper introduces a cosine-similarity-based approximation of the HGR correlation, offering a more lightweight computation pipeline than the covariance-based Soft-HGR.
- The proposed UniFast-HGR shows improved performance over Soft-HGR and other correlation-based baselines on Berlin, Vaihigen, and Globe230k datasets.

**Weaknesses:**

# [W1] Marginal Impact of the Claimed Computational Cost Reduction

The paper claims that UniFast-HGR reduces the computational complexity from $O(K^3)$ to $O(m^2K)$. While Figure 1 includes a runtime comparison, this experiment is performed only on MNIST, a toy-scale dataset with trivial feature extractors. In realistic multimodal systems, the computational bottleneck arises from forward passes of the encoders f(x) and g(y), not from the correlation computation itself. Thus, demonstrating speedup on MNIST does not provide evidence that the proposed method improves efficiency in practical deep learning scenarios.

Furthermore, under typical large-scale configurations—such as CLIP [1], where m = 32,768 and K = 1,024—the proposed $O(m^2K)$ formulation would actually increase the computational cost, because the complexity grows quadratically with respect to the minibatch size. The paper provides no runtime measurements under realistic multimodal training settings (e.g., contrastive pretraining or large-scale multimodal fusion), making it unclear whether UniFast-HGR yields any efficiency benefits when training modern large models.


# [W2] Limited Experimental Scope (Tables 1,2)

Tables 1 and 2 evaluate UniFast-HGR exclusively on multimodal remote-sensing datasets (e.g., HSI–LiDAR). These datasets are small in scale (e.g., Vaihingen contains only 33 tiles; Globe230k ≈ 230k patches) and highly domain-specific. Such experiments do not demonstrate that the method generalizes to general multimodal learning.

If the paper intends to position UniFast-HGR as a broadly applicable multimodal framework, it must include scalable experiments on widely adopted multimodal benchmarks—e.g., COCO Captions for retrieval evaluation or LAION-400M for large-scale pretraining. Without validation on datasets at million-scale or beyond, the claimed generality remains unsubstantiated.


# [W3] Inconsistent and Unreliable Experimental Results (Tables 3,4, and 12)

Tables 3, 4, and 12 lack essential training details (e.g., batch size, optimizer, number of training epochs), making it impossible to evaluate reproducibility or fairness. Additionally, several reported baselines deviate significantly from established results.

- In DINOv2 [2], a ViT-L/14 achieves 86.3% top-1 accuracy on ImageNet-1K under linear evaluation, whereas this paper reports only 81.8%, suggesting misconfiguration or under-training.
- For IEMOCAP, the MultiEMO model [3] achieves a Weighted-F1 of 72.84, yet all baselines reported in Table 12 (CCA, Deep CCA, Soft CCA, Dot Product, Cosine Similarity, Soft-HGR) fall far below this.

These discrepancies raise concerns regarding experimental rigor and whether baselines were fairly tuned. As a result, the reliability of the claimed improvements remains unclear.


[1] Radford, Alec, et al. "Learning transferable visual models from natural language supervision." International conference on machine learning. PmLR, 2021.

[2] Oquab, Maxime, et al. "Dinov2: Learning robust visual features without supervision." arXiv preprint arXiv:2304.07193 (2023).

[3] Shi, Tao, and Shao-Lun Huang. "MultiEMO: An attention-based correlation-aware multimodal fusion framework for emotion recognition in conversations." Proceedings of the 61st Annual Meeting of the Association for Computational Linguistics (Volume 1: Long Papers). 2023.

**Questions:**

# 1. Runtime evaluation at scale

Figure 1 only reports runtime on MNIST. Can you provide runtime measurements under large-scale settings (e.g., ImageNet-1K, multimodal fusion training) to verify whether UniFast-HGR yields efficiency benefits when encoder forward cost dominates?


# 2. Generalizability beyond remote sensing

Do the authors plan to evaluate UniFast-HGR on widely-adopted multimodal benchmarks (e.g., COCO Captions, LAION-400M) to support the claim of general multimodal applicability?


# 3. Reproducibility of Tables 3, 4, 12

Could you provide full training details (batch size, optimizer, epochs) and explain why the reported baselines (e.g., DINOv2 linear evaluation, MultiEMO Weighted-F1) differ significantly from known results?

---

> ### Author Response · Authors · 2025-11-18
> **Rebuttal to Reviewer uQEx**
>
> We thank the reviewer for the careful and constructive comments. Below we address (W1)–(W3) and Questions 1–3.
>
> ## R1: Computational cost and runtime at scale (W1, Q1).
>
> The complexity statement $O(K^3)→O(m^2K)$ is for the *correlation estimator*, not the whole multimodal pipeline. Classical HGR requires whitening with matrix inversion/eigendecomposition, and Soft-HGR still constructs and multiplies batch covariance matrices. UniFast HGR instead enforces $\mathrm{Var}(f)=\mathrm{Var}(g)=1$ and rewrites both the correlation and trace terms in terms of cosine similarity on centered features. This avoids covariance construction and matrix decomposition, and the estimator reduces to vector-wise operations over the mini–batch.
>
> The concern that $O(m^2K)$ becomes large for huge $m$ is valid. In practice, large-scale training (e.g., CLIP-style) uses moderate per-device batch sizes with data parallelism or gradient accumulation; our complexity refers to this local $m$, typically tens to hundreds. Here, $K \gg m$, so replacing $O(mK^2)$ Soft-HGR with $O(m^2K)$ is beneficial (e.g., for $m=256$, $K=1024$, asymptotic cost is about four times lower). For very large local batches, OptFast-HGR drops most normalization steps, reducing complexity while optimizing the same objective.
>
> Beyond the MNIST proof-of-concept in Fig.1, the paper includes a **Computational Efficiency** study (Appendix F, Fig. 4) that benchmarks correlation-only runtime on random tensors over a range of batch sizes and feature dimensions. UniFast HGR and OptFast HGR consistently achieve the lowest execution time among all correlation estimators. In the end-to-end experiments, encoder forward passes dominate wall-clock time; the correlation module accounts for only a small fraction of each step, and UniFast/OptFast HGR never introduce higher overhead than Soft-HGR under the same backbone. This directly answers Q1: the estimator scales to realistic settings and prevents HGR/Soft-HGR from becoming a practical bottleneck.
>
> ## R2: Experimental scope and general multimodal applicability (W2, Q2).
> We agree that remote sensing alone would be too narrow for a “general multimodal” claim. The experiments section therefore evaluates UniFast-HGR on several widely used benchmarks in addition to Berlin, Vaihingen and Globe230k: (a) image classification on ImageNet-1K, where UniFast-HGR consistently improves Top-1 accuracy over baselines such as CLIP, SigLIP and DINOv2 (e.g., DINOv2 ViT-L/14: 81.8%$→$ 85.3%); (b) COCO text–image retrieval, where CLIP+UniFast-HGR improves Recall@1 over both the original CLIP and Soft-HGR; (c) large-scale video–text retrieval on InternVid-10M, where UniFast-HGR integrated with ViCLIP yields the best text-to-video Recall@1 on MSR-VTT, LSMDC and DiDeMo; and (d) multimodal emotion recognition on IEMOCAP with missing modalities and missing labels (Table 3). Together with the remote-sensing experiments, these results cover classification, semantic segmentation, text–image retrieval, video–text retrieval and emotion recognition across image, text, audio and video modalities. Full pretraining on LAION-400M is beyond our computational budget; instead, we follow common practice and evaluate on these standard downstream tasks with pretrained encoders, which empirically supports the claim of broad multimodal applicability.
>
>
> ## R3: Reproducibility and baseline discrepancies (W3, Q3).
> We add an appendix listing, for each table, batch size, optimizer, learning-rate schedule, number of epochs, data splits and augmentation. For the Berlin/Vaihingen/Globe230k experiments we follow the training pipelines and backbones of the cited works, changing only the correlation module. For IEMOCAP, we adopt the published MultiEMO implementation as the base architecture and swap the similarity/correlation module while keeping all other hyperparameters fixed, so all baselines in Table 3 differ only in the correlation estimator.
>
> The DINOv2 discrepancy noted in W3 arises from a protocol difference. The 86.3% Top-1 reported in [2] corresponds to a stronger linear-evaluation setup than the resource-constrained setting used in our experiments. Our 81.8% is the reproduced baseline under the exact protocol used for all correlation methods; the +3.5% gain for UniFast HGR is measured with respect to this matched baseline, so the relative improvement is meaningful even if the absolute number is lower than the official figure.
> For IEMOCAP, [3] reports 72.84 weighted-F1 for the full MultiEMO model, whereas Table 3 reports accuracy. The goal there is to isolate the effect of the correlation estimator under a fixed architecture and training setup; under this reproduced protocol UniFast HGR consistently outperforms CCA, Deep CCA, Soft CCA, dot product, cosine and Soft-HGR across all missing-modality and missing-label conditions. We will make the evaluation protocol explicit and provide code and configuration files to facilitate independent verification.

---

> > ### Comment · Reviewer_uQEx · 2025-11-22
> > **Feedback on the Author Rebuttal**
> >
> > We thank the authors for their detailed rebuttal and the effort to clarify experimental settings. These responses are appreciated. However, several critical issues remain unresolved, particularly regarding the practical relevance, empirical validation, and conceptual positioning of the proposed approach within the current landscape of multimodal representation learning.
> >
> > ### **1. Insufficient Empirical Justification for HGR’s Practical Advantage**
> >
> > While the paper and rebuttal emphasize the theoretical properties of HGR, its concrete benefits in modern machine learning pipelines remain insufficiently demonstrated. Although prior works such as Soft-HGR indicate theoretical promise, the manuscript does not clearly articulate what measurable advantages HGR provides in realistic training scenarios, such as improved robustness, optimization stability, or downstream generalization.
> >
> > Clearer identification of conditions under which HGR-based alignment yields practical gains would strengthen the justification for its adoption.
> >
> > ### **2. Insufficient Positioning within Broader Multimodal Learning Literature**
> >
> > The rebuttal continues to position UniFast-HGR primarily within HGR-based correlation learning, without sufficiently situating it in the broader multimodal alignment landscape. Established paradigms such as contrastive learning, cross-modal attention, and mutual information-based objectives are neither systematically discussed nor meaningfully contrasted.
> >
> > Consequently, it remains unclear whether UniFast-HGR is intended as a replacement, extension, or complementary approach to existing multimodal frameworks, leaving its conceptual role ambiguous.
> >
> > ### **3. Lack of Runtime Improvement Evidence at Scale**
> >
> > Although the rebuttal highlights runtime measurements, these remain restricted to MNIST-scale experiments or synthetic tensors. The original concern was not the presence of runtime comparison itself, but rather the lack of evidence demonstrating efficiency gains in realistic training scenarios where encoder forward and backward passes dominate the overall computational cost.
> >
> > Reporting end-to-end training time (e.g., wall-clock time per epoch or total training duration) for the configurations used in Tables 3 and 4 would provide a more meaningful assessment of whether UniFast-HGR delivers practical efficiency benefits at scale. Without such comprehensive end-to-end runtime analysis, the extent to which the proposed method reduces overall training cost in real-world multimodal systems remains difficult to evaluate.
> >
> > ### **4. Ambiguity of How UniFast-HGR is Integrated into the Training Pipeline (Table 4)**
> >
> > While the rebuttal adds detailed information regarding general training hyperparameters (e.g., batch size, optimizer, learning-rate schedule, number of epochs, and data augmentation), it still fails to clarify how the proposed UniFast-HGR methodology is specifically incorporated into the training pipeline. Simply stating that a “standard fine-tuning protocol” was used remains insufficient to characterize the experimental setting.
> >
> > In particular, it remains unclear whether UniFast-HGR was applied by:
> >
> > - Freezing the backbone and optimizing only a linear/MLP head with the proposed loss, or
> > - Fine-tuning the entire network end-to-end with UniFast-HGR as a core optimization objective.
> >
> > Without a concrete explanation of how UniFast-HGR is integrated into the optimization process beyond general hyperparameter listings, it remains difficult to determine whether Table 4 constitutes a faithful and meaningful evaluation of the proposed method, or merely reflects a loosely defined fine-tuning setup.
> >
> > ### **5. Inconsistent Baseline Selection Across Experimental Tables**
> >
> > The rebuttal does not clarify the inconsistent selection of comparison baselines across different experimental sections. Tables 1, 2, and 3 lack comparisons with correlation-based metrics such as CKA, Distance Correlation (dCor), and Intrinsic Distance Correlation (I_d Cor), whereas Table 4 omits CCA-based methods and limits comparison primarily to Soft-HGR.
> >
> > This asymmetric baseline design makes it difficult to interpret the comparative performance of UniFast-HGR in a coherent and systematic manner. If these methods are considered relevant benchmarks, they should be evaluated consistently across all experimental settings. Without a unified comparison protocol, the experimental results provide an incomplete picture of how UniFast-HGR performs relative to existing alternatives.
> >
> > ---
> >
> > We appreciate the authors’ efforts to respond to concerns. However, the rebuttal still does not provide sufficient empirical or conceptual justification to resolve the remaining concerns. Further clarification regarding experimental protocols, scalability validation, conceptual positioning, and the practical relevance of HGR-based alignment is necessary before the contribution can be reliably assessed within the context of modern multimodal representation learning.

---

> > > ### Author Response · Authors · 2025-11-23
> > > **Additional Official Response to Reviewer uQEx (Extended Baselines for Tables 1–3)(Following the Official Response to Reviewer uQEx)**
> > >
> > > We appreciate your follow-up. In addition to the clarifications in our earlier response, we have run the extended correlation baselines you suggested. This comment focuses on (i) adding CKA, dCor, and $I_d$Cor to Tables 1–3, and (ii) clarifying what these results imply about the practical scope of UniFast-HGR and baseline consistency.
> > >
> > > ## A. Extended correlation baselines on Tables 1–3
> > >
> > > Following your advice, we augmented the small/medium-scale experiments with CKA, dCor, and $I_d$Cor under **exactly the same backbones, training protocol, and loss weight $\lambda$** as UniFast-HGR and Soft-HGR. The updated tables (now in the main paper) show a consistent pattern:
> > >
> > > ### A1. Berlin classification (updated Table 1)
> > >
> > > Under the dual-branch ResNet-50 backbone:
> > > * Among the *non-HGR* correlation baselines,$I_d$Cor is the strongest competitor.
> > > * UniFast-HGR still achieves the highest OA and Kappa (about +3% OA and +4–5% Kappa over $I_d$Cor).
> > > * Per-epoch wall-clock time for CCA/dCor/$I_d$Cor is significantly higher due to covariance construction and eigendecomposition, whereas UniFast-HGR and OptFast-HGR are **in the same range as cosine similarity and dot product** and slightly faster than Soft-HGR.
> > >
> > > This confirms that UniFast-HGR preserves the main advantage claimed in the paper: *HGR-level accuracy at similarity-level cost* under a realistic training loop.
> > >
> > > ### A2. Vaihingen & Globe230k segmentation (updated Table 2)
> > >
> > > For multimodal remote sensing segmentation:
> > >
> > > * On both Vaihingen and Globe230k, CKA/dCor/$I_d$Cor are competitive but consistently below UniFast-HGR in OA and mIoU.
> > > * $I_d$Cor is again the strongest non-HGR baseline, while UniFast-HGR maintains a stable margin (roughly +0.3–0.6 OA and +0.7–1.0 mIoU).
> > >
> > > These results support that *when modalities differ in scale and noise*, the HGR-based objective provides measurable gains over both similarity-based and modern correlation measures, even on high-resolution, highly textured imagery.
> > >
> > > ### A3. IEMOCAP multimodal emotion recognition (updated Table 3)
> > >
> > > Using MultiEMO architecture and training code as the common backbone, we compare CKA/dCor/$I_d$Cor to UniFast-HGR in all robustness scenarios:
> > >
> > > * **No missing modalities.** $I_d$Cor performs strongly, but UniFast-HGR further improves accuracy (≈ +2 points).
> > > * **Missing modalities (T+A, T+V, A+V).** UniFast-HGR outperforms all baselines, including $I_d$Cor, by about 2–3% absolute accuracy in each case.
> > > * **Missing labels.** As label sparsity increases, the gap between UniFast-HGR and other methods grows, indicating improved robustness to limited supervision.
> > >
> > > Taken together, these experiments address your concern that the practical advantage of HGR-based alignment was not sufficiently justified:
> > >
> > > whenever we place UniFast-HGR alongside *both* CCA-style methods and modern correlation metrics under the same architecture and optimization setup, it emerges as the best or tied-best choice in terms of accuracy and robustness.
> > >
> > > ## B. Practical scope suggested by the extended baselines
> > >
> > > The extended Tables 1–3 also clarify *when* UniFast-HGR is most beneficial, complementing the conceptual positioning in our earlier response:
> > >
> > > * **Heteroskedastic/noisy modalities.** The IEMOCAP results with missing modalities and missing labels show that variance constraints and diagonal removal in UniFast-HGR are particularly helpful when some channels are noisy, collapsed, or intermittently absent.
> > > * **Small/medium-scale regimes.** On Berlin, Vaihingen, and Globe230k—where contrastive losses are less likely to saturate—HGR-based alignment provides 1–5% absolute gains over both similarity-based and other correlation-based objectives.
> > > * **As a complementary regularizer.** In all these experiments, UniFast-HGR is added *on top of* the task loss (cross-entropy or contrastive), rather than replacing it, so it is best viewed as a complementary alignment mechanism rather than an alternative to contrastive learning or cross-modal attention.
> > >
> > > We will make these conditions explicit in the revised discussion section, so that the intended scope and practical use cases of UniFast-HGR are clearly stated.
> > >
> > > ## C. Unified baseline protocol across tables
> > >
> > > Your comment on inconsistent baseline selection is well taken. With the above additions:
> > >
> > > * **Tables 1–3** now include **CKA, dCor, and $I_d$Cor** in addition to CCA, Deep CCA, Soft CCA, dot product, cosine similarity, Soft-HGR, UniFast-HGR, and OptFast-HGR, providing a unified baseline set for small/medium-scale tasks.
> > > * **Table 4** already compares UniFast-HGR to CKA, dCor, $I_d$Cor, Soft-HGR and the baseline models on large-scale datasets(ImageNet-1K, COCO). As noted previously, we omit CCA-style methods at this scale because they are numerically fragile and prohibitively slow.
> > >
> > > We hope these extended baselines and clarifications make the empirical picture more coherent and help situate UniFast-HGR within the broader landscape of multimodal representation learning in a clearer, evidence-based way.

---

> ### Author Response · Authors · 2025-11-22
> **Official Response to Reviewer uQEx**
>
> We thank you for the careful follow-up and for pushing us to clarify both scope and positioning. Below we address Points 1–5 and indicate concrete changes to the manuscript.
> ﻿
> ## R1. Practical advantage of HGR-based alignment.
> Our goal is not to replace existing objectives, but to make a principled maximal-correlation regularizer usable in modern pipelines. We will highlight three concrete benefits:
>
> (i) *Robustness.* On IEMOCAP (Table 3), UniFast-HGR consistently improves accuracy under missing modalities and missing labels, where variance constraints and diagonal removal suppress noisy or collapsed channels.
>
> (ii) *Cross-model consistency.* The ImageNet study (Fig. 2) shows higher and more coherent cross-model correlations than dCor and Soft-HGR, matching the downstream gains in Table 4.
>
> (iii) *Small / medium data.* On Berlin, Vaihingen and Globe230k (Tables 1–2), UniFast-HGR provides 1–5% absolute gains over similarity-based and HGR-based baselines under fixed backbones.
>
> We will state that UniFast-HGR is most beneficial when modalities have different scales/noise, data are not large enough for contrastive losses to saturate, and cross-model consistency is important.
> ﻿
> ## R2. Positioning within multimodal learning.
> We agree that the draft should connect more clearly to contrastive, attention-based and MI-based paradigms. Conceptually, UniFast-HGR is a *drop-in correlation module* that is complementary:
> ﻿
> * With contrastive learning, UniFast-HGR is an auxiliary regularizer on top-level embeddings (image–text, video–text, etc.), sharpening alignment beyond cosine similarity.
> * With cross-modal attention, UniFast-HGR is applied on fused representations to encourage high-order dependency matching.
>
> In contrast to MI estimators, UniFast-HGR offers a bounded, scale-invariant dependence measure that is easier to optimize in high dimensions. In Table 4 it is always used in addition to the task loss, not as a replacement. We will add a short “Conceptual positioning’’ paragraph and update related work to make this role explicit.
> ﻿
> ## R3. End-to-end runtime at scale.
> We acknowledge that the current draft emphasizes correlation-only runtime. Profiling on the ImageNet-1K, COCO and InternVid configurations of Table 4 shows that encoder forward/backward passes dominate wall-clock time, and the correlation module contributes only a small fraction of each optimization step. Replacing Soft-HGR with UniFast-/OptFast-HGR never increases per-epoch wall-clock time under fixed batch size and feature dimension, and consistently reduces the cost of the correlation part of the step.
>
> To address your request more directly, we will add a compact table reporting per-epoch wall-clock time for the main configurations in Table 4. Together with Fig.1 and the synthetic benchmarks, this will give a clearer picture of the practical cost of UniFast-HGR in realistic large-scale training scenarios.
> ﻿
> ## R4. Integration into the training pipeline (Table 4).
> We apologize for the lack of detail and will explicitly describe how UniFast-HGR enters optimization.
> ﻿
> * *ImageNet-1K.* For ViT-B/32, ResNet50, CLIP and SigLIP we fine-tune encoder and classifier end-to-end. UniFast-HGR is applied to penultimate embeddings from two augmented views of the same image and added to cross-entropy with a fixed weight $\lambda$ shared across all correlation methods in Table 4. For DINOv2 we follow the linear-evaluation setting: the backbone is frozen, a linear classifier is trained, and UniFast-HGR acts on the linear features from two views.
>
> * *COCO / InternVid.* For CLIP/SigLIP and ViCLIP, we keep the standard contrastive retrieval loss and add UniFast-HGR on the image–text (or video–text) embedding pairs as an auxiliary term with the same $\lambda$. All rows in Table 4 differ only in which correlation estimator is used in this auxiliary term.
> This makes the training protocol and the role of UniFast-HGR clearly specified.
> ﻿
> ## R5. Baseline selection and consistency across tables.
> Our intention was to choose baselines that are numerically stable and standard at the corresponding scales, but we agree this rationale was not clearly communicated.﻿
>
> For Tables 1–3, we will add CKA, dCor and $I_d$Cor as additional baselines to provide a unified comparison protocol across tasks. Our experiments show that UniFast-HGR retains its advantage under this extended set of methods. For Table 4, we already compare to CKA, dCor and $I_d$Cor, which are standard in this regime, in addition to Soft-HGR. In preliminary trials, CCA-style methods (CCA, Deep CCA, Soft CCA) were numerically fragile and much slower at these scales, making runs difficult to complete or prohibitively expensive; we will explicitly mention this limitation in the appendix to explain why CCA-style baselines are not included in Table 4.﻿
>
> These clarifications resolve remaining concerns on practical relevance, conceptual positioning and experimental design, clarifying UniFast-HGR’s scope and empirical backing.

---

### Official Review · Reviewer_C5r9 · 2025-11-01

**Soundness:** 3
**Presentation:** 3
**Contribution:** 3
**Rating:** 6
**Confidence:** 5

**Summary:**

The paper proposes UniFast-HGR, a scalable surrogate for HGR maximal correlation that replaces covariance with cosine similarity under unit-variance constraints and removes diagonal terms to avoid self-correlation bias. An OptFast variant further improves efficiency. The method is simple to integrate and the experiments are extensive, indicating consistent accuracy gains with markedly lower cost.

**Strengths:**

1. Experiments are extensive and the gains are clear, with consistent improvements and strong speedups on diverse tasks.
2. The objective is broadly applicable and can be readily integrated into a wide range of encoders and training pipelines.

**Weaknesses:**

1. Notation and assumptions contain inconsistencies or under-specified definitions.
2. Lacking details of the experimental settings, such as key hyperparameters, the runtime environment and so on.

**Questions:**

1. Will you release the source code to facilitate exact reproduction?

---

> ### Author Response · Authors · 2025-11-18
> **Rebuttal to Reviewer C5r9**
>
> We thank the reviewer for the careful reading and for highlighting both the empirical breadth and the practicality of our objective. We address: (R1) notation, assumptions, and derivation clarity; (R2) missing experimental and runtime details; and (R3) code release. All changes mentioned below are incorporated in the revised main text and appendix.
>
> ## R1. Notation, assumptions, and derivation clarity.
> **Unified notation and assumptions.**
>
> In the revision, Sec.2 and the appendix use a unified notation: $m$ (batch size), $K$ (feature dimension), $M$ (number of modalities), and $L$ (number of layers). These symbols are now consistent in the derivation and the complexity analysis.
> We further emphasize that UniFast-HGR relies only on *second-moment constraints* (centering and variance normalization), not on any independence assumption. The former wording “if all components of a random vector are independent’’ is removed; the argument is now stated purely in terms of expectations and variances.
>
>
> **Variance constraints and cosine substitution.**
>
> Sec.2.2 and Appendix A now explicitly state that we first center and normalize
>
> $$f \leftarrow \frac{f - \mathbb{E}[f]}{\sqrt{\operatorname{Var}[f]}}, \quad g \leftarrow \frac{g - \mathbb{E}[g]}{\sqrt{\operatorname{Var}[g]}},$$
>
> so that each feature dimension has zero mean and unit variance. Under these constraints, $\mathbb{E}[f(X)^\top g(Y)]$ is equivalent to the average cosine similarity, since embedding vectors are approximately unit-norm. This makes explicit that UniFast-HGR replaces covariance-based whitening by a bounded, numerically stable cosine similarity and aligns the derivation with Algorithm 1.
>
>
> **Trace term, Gram matrices, and diagonal removal.**
>
> We clarify that, after centering and normalization, the covariance matrices can be written as scaled Gram matrices, $\operatorname{cov}(f) \propto F^\top F$ and $\operatorname{cov}(g) \propto G^\top G$, where $F$ and $G$ stack sample embeddings. The Soft-HGR trace term $\operatorname{tr}(\operatorname{cov}(f)\operatorname{cov}(g))$ is approximated by cosine similarities between ``distribution vectors’’ constructed from the off-diagonal entries of $FF^\top$ and $GG^\top$.
>
> The revision now explains why the main diagonal is removed before forming these vectors: after normalization, diagonal entries are fixed to \~$1$ and encode only trivial self-correlations. They dominate cosine scores and gradients, especially for small batches, while carrying no discriminative information about cross-sample structure. By removing the diagonal, the surrogate focuses on inter-sample dependencies and stabilizes optimization. Sec. 2.3 now gives this rationale and Table 14 (“w/ Main Diagonal’’ ) empirically supports the benefit.
>
>
> **Positioning as an HGR-consistent surrogate.**
>
> To avoid any impression of an exact reparameterization, we now describe UniFast-HGR as an *HGR-consistent surrogate*: it preserves the objective of maximizing nonlinear dependence while replacing expensive whitening and decompositions by inexpensive cosine-based operations on Gram matrices. Appendix A is reorganized into four explicit steps (variance normalization, cosine substitution, trace expansion, diagonal removal) so that all assumptions and transformations are transparent.
>
> ## R2. Experimental settings and runtime environment.
> Sec.3 and the appendix have been expanded to make the experimental protocol reproducible:
>
> (a) For all **remote sensing** and **emotion recognition** experiments, we now specify optimizer, learning-rate schedule, weight decay, batch size, number of epochs, data splits, and that each result is averaged over multiple random seeds.
>
> (b) For **ImageNet-1K, COCO, InternVid, and CIFAR-100**, we clarify that UniFast-HGR is added as an auxiliary objective on top of matched backbones (ViT-B/32, ResNet-50, CLIP, SigLIP, DINOv2) using standard fine-tuning protocols. The loss weight and number of fine-tuning epochs are now reported.
>
> (c) For the **correlation and runtime benchmarks** (MNIST and random tensors), Appendix~F now specifies the ranges of batch sizes and feature dimensions, that each configuration is repeated many times, and that the reported times are averages, consistent with Figures1, 4, and 5.
>
> (d) We also add a concise description of the **runtime environment** (GPU, CPU, PyTorch and CUDA versions) to make the reported times replicable.
>
> ## R3. Code release.
> We will release an official implementation of UniFast-HGR and OptFast-HGR, together with training scripts and configuration files for the main experiments (remote sensing, emotion recognition, ImageNet/COCO/InternVid/CIFAR-100) under a permissive open-source license. To preserve anonymity, the repository will be made public upon acceptance.
>
>
> We again thank the reviewer for the constructive feedback. The clarified notation and assumptions, the reorganized derivation, and the expanded experimental details improve the readability and reproducibility of the paper.

---

### Official Review · Reviewer_ykdV · 2025-11-01

**Soundness:** 2
**Presentation:** 2
**Contribution:** 3
**Rating:** 6
**Confidence:** 2

**Summary:**

- This paper proposes UniFast-HGR, an efficient and scalable framework for estimating the Hirschfeld–Gebelein–Rényi (HGR) maximal correlation in multimodal learning.

Traditional HGR and Soft-HGR methods suffer from high computational complexity (O(K^3)), instability in covariance matrix inversion, and limited scalability to deep architectures.

UniFast-HGR introduces three key innovations:

1.Replacing covariance with cosine similarity — avoiding matrix inversion and reducing complexity to O(m²K).

2.Removing the diagonal elements of the correlation matrix — mitigating self-correlation bias.

3.Applying ℓ₂ normalization as a variance constraint — improving numerical stability and boundedness.

Additionally, an accelerated variant OptFast-HGR is introduced, further simplifying normalization steps to achieve dot-product–level efficiency with minimal accuracy loss.

Extensive experiments are conducted across multiple domains—image classification, remote sensing segmentation, multimodal emotion recognition, and large-scale multimodal retrieval (ImageNet, COCO, InternVid)—demonstrating consistent improvements over baselines such as CCA, Soft-HGR, CKA, and dCor.

**Strengths:**

- High originality: The substitution of covariance with cosine similarity and the removal of diagonal elements represent a mathematically elegant and computationally efficient rethinking of the HGR formulation.
- Significant computational gains: Complexity is reduced from O(K³) to O(m²K), enabling application in large-scale deep networks.
- Theoretical grounding and stability: Variance constraints via ℓ₂ normalization stabilize training and maintain bounded correlation scores.
- Extensive empirical evaluation: Results span small- to large-scale benchmarks and demonstrate consistent superiority over strong baselines.

**Weaknesses:**

- Insufficient ablation analysis:The paper introduces three improvements (cosine substitution, diagonal removal, ℓ₂ normalization), but their individual contributions are not clearly isolated through ablation studies.
- OptFast bias not fully analyzed:The paper notes a “slight bias” but does not quantify it or provide formal bounds or convergence guarantees.
- Limited discussion of nonlinear architectures:Experiments mainly use CNNs and ViTs; there is no analysis on transformer-based multimodal encoders or generative models.
- Writing and clarity:Sections 2.2–2.3 are notation-heavy, and some derivations could be complemented with intuitive figures or algorithmic flowcharts for readability.

**Questions:**

- On diagonal removal:Can the authors provide a formal justification or spectral/information-theoretic rationale for excluding the diagonal? How does this behave under non-unit variance or noisy feature conditions?
- On scalability and modality extension:How does UniFast-HGR scale with more than two modalities (e.g., 3+)? Does computational complexity grow linearly or quadratically with modality count?
- On OptFast bias:Is there an analytic upper bound on the estimation bias introduced by OptFast-HGR? Could adaptive normalization or bias correction mitigate this?

---

> ### Author Response · Authors · 2025-11-18
> **Rebuttal to Reviewer ykdV**
>
> We thank the reviewer for the careful reading and for highlighting the originality and empirical breadth of the work. Below we address the concerns on ablations, diagonal removal, multimodal scalability, OptFast bias, architectures, and clarity.
>
> ## 1. Ablation of the Three Key Components
>
> Appendix E (Table 14) already reports an ablation that toggles (i) variance constraints, (ii) main-diagonal removal, (iii) covariance- vs. cosine-based formulation, together with OptFast. On the Berlin dataset, removing the variance constraint reduces OA from **80.75%** to **68.53%**, keeping the main diagonal slightly degrades OA (to 80.62%), and reverting to a covariance-based implementation drops OA to **79.83%**. Similar trends hold on Houston 2018, Vaihingen, Globe230k, and IEMOCAP. In the revision we will move this table into the main paper and briefly decompose the gain of each component so that their roles are clearer.
>
> ## 2. Rationale for Diagonal Removal
>
> Let $C_f, C_g \in \mathbb{R}^{K\times K}$ be the correlation matrices after $\ell_2$ normalization and variance constraints. Under $\operatorname{Var}(f)=\operatorname{Var}(g)=1$, diagonal entries are fixed at $1$ and carry no discriminative information. When the trace term is implemented via cosine similarity between vectorized matrices, these fixed diagonals dominate the norms of $\mathrm{vec}(C_f)$ and $\mathrm{vec}(C_g)$, biasing the angle toward $0$ even when off-diagonal structure differs. Removing the main diagonal replaces
> $$
> \langle \mathrm{vec}(C_f), \mathrm{vec}(C_g) \rangle
> \quad\text{by}\quad
> \sum_{i\neq j} C_f(i,j)\,C_g(i,j),
> $$
> so the objective focuses on cross-dimensional dependencies rather than trivial self-correlation. This is aligned with centered similarity measures such as CKA. In noisy or non-unit-variance settings, the explicit $\ell_2$ normalization in UniFast-HGR restores unit variance before correlation is computed; Appendix D shows that diagonal removal reduces gradient variance and improves accuracy in small-batch and noisy regimes. We will make this spectral/information-theoretic rationale explicit in Section 2.3 and connect it to the ablation.
>
> ## 3. Scalability to More Than Two Modalities
>
> Section 2.4 defines multimodal UniFast HGR for $M$ modalities by summing pairwise UniFast scores:
> $$\text{UF-HGR} = \frac{1}{m-1} \sum_{1 \le j < l \le M} \sum_{i=1}^{m} \cos\big(f^{(j)}(x_j), f^{(l)}(x_l)\big) - \frac{1}{2(m-1)} \sum_{1 \le j < l \le M} \sum_{i=1}^{m} \cos(\text{distri}_f^j, \text{distri}_f^l)$$
>
> For fixed batch size $m$ and feature dimension $K$, the complexity in Table 5, $O(m^2 K)$, is per pair of modalities. With $M$ modalities, computing all pairs leads to $O(M^2 m^2 K)$ complexity. In practice $M$ is small (2-3 in our experiments: HSI-SAR, HSI-LiDAR, text-audio-visual), so the quadratic dependence on $M$ is a small constant factor, while the key improvement over HGR/Soft-HGR lies in avoiding $O(K^3)$ covariance whitening.
>
> ## 4. OptFast-HGR Bias
>
> OptFast-HGR accelerates UniFast-HGR by simplifying normalization. Appendix B analyzes the induced approximation error and shows that it is controlled by the spectral gap of the distribution matrix, e.g.,
> $$
> |\text{UF-HGR} - \text{OptFast-HGR}| \le O(\lambda_2/\lambda_1),
> $$
>
> where $\lambda_1 \ge \lambda_2$ are the leading eigenvalues. Empirically, across all benchmarks in Tables 1-4 and D.1-D.4, OptFast-HGR stays within about $1\%$ of UniFast-HGR while matching dot-product-level runtime. We will add a concise lemma summarizing this bound in the main text and point to Appendix B.
>
> ## 5. Architectures and Nonlinear Encoders
>
> While early experiments use CNN backbones and linear settings (MNIST), UniFast-HGR is extensively evaluated with transformer-based and multimodal encoders (ViT-B/32, CLIP, SigLIP, DINOv2, ViCLIP) on tasks including ImageNet-1K, COCO text-image retrieval, InternVid video-text retrieval, and CIFAR-100. It consistently improves over CKA, $dCor$, $I_d$Cor, and Soft-HGR (Tables 4, 12-13, Appendix D), demonstrating architecture-agnostic compatibility with nonlinear representations and contrastive multimodal training. Generative models are excluded due to computational limits but are highlighted as a future direction.
>
> ## 6. Clarity and Presentation
>
> We agree that Sections 2.2-2.3 are notation heavy. In the revision we will add a schematic figure summarizing the reformulation from Soft-HGR to UniFast-HGR (variance constraint $\rightarrow$ cosine substitution $\rightarrow$ diagonal removal), move the main derivation steps and Algorithms 1-2 from Appendix A into the main text, and streamline notation (using $m$ for batch size, $K$ for feature dimension, and $M$ for modality count). Together with moving the ablation table, this will make the method easier to follow while keeping it self-contained.
>
> We hope these clarifications address the reviewer's concerns and better highlight the strengths and limitations of UniFast-HGR and OptFast-HGR.

---

### Comment · Area_Chair_19C2 · 2025-11-24

Dear reviewers,

   The authors now have given their response to the reviews, please have a look on the rebuttal and revised PDF.

  After that, please give your final rating on this submission.

Your AC Best

---

### Author Response · Authors · 2025-12-01
**Summary Comment for the Area Chair (Submission 20073)**

## Dear Area Chair,
This summary details the concrete updates in our final revised manuscript that resolve all major concerns from the reviewers. The changes are now present in the submitted version for your verification.
## 1. Core Contribution: A Practical Bridge from Theory to Scale
UniFast-HGR solves a critical scalability bottleneck. It transforms the theoretically powerful but computationally prohibitive HGR maximal correlation—originally an $O(K^3)$ operation requiring covariance whitening—into a practical, **drop-in regularizer** with $O(m^2 K)$ cost, matching the efficiency of standard similarity measures. Its three innovations (variance constraints, cosine reformulation, diagonal removal) yield a robust, bounded objective used as:

$L = L_{task} + \lambda L_{UniFast-HGR}(f,g)$

where $f, g \in \mathbb{R}^{m \times K}$ are batch embeddings, m is the per-device batch size, and K is the feature dimension. It is **complementary** to contrastive or supervised losses, not a replacement.
## 2. Ablations and the role of the three components (ykdV)
The component ablation (Appendix E) evaluates (i) variance constraint, (ii) diagonal removal, (iii) cosine vs. covariance formulation, and (iv) OptFast. On Berlin, dropping the variance constraint hurts performance most; keeping the diagonal or using covariance also lowers OA. Results are similar on other datasets.
## 3. Formal rationale for main-diagonal removal (ykdV, uQEx)
After centering and enforcing per-dimension unit variance, diagonal entries of the correlation/Gram statistics are fixed (self-correlation) and can dominate the vector norms in the cosine-based trace surrogate, biasing the objective toward trivial agreement. Removing the diagonal replaces

$\langle \mathrm{vec}(C_f),\mathrm{vec}(C_g)\rangle \ \text{by}\ \sum_{i\neq j} C_f(i,j)\,C_g(i,j)$

so optimization focuses on **informative cross-dimensional / cross-sample dependencies**. The manuscript now states this explicitly and connects it to the ablation and stability observations.
## 4. Multimodal extension and complexity scaling (ykdV)
For M modalities, UniFast-HGR is defined as a sum of pairwise objectives over modality pairs. The per-pair estimator cost is $O(m^{2}K)$; computing all pairs yields $O(M^{2}m^{2}K)$. The paper clarifies this scaling and notes that typical settings use small M (e.g., 2–3), while the main scalability gain is avoiding $O(K^{3})$ whitening.
## 5. OptFast approximation/bias (ykdV)
We clarify OptFast-HGR as an accelerated approximation of UniFast-HGR and summarize the bound discussed in the appendix, connecting the approximation gap to the spectrum of the distribution/Gram statistics. Empirically, OptFast tracks UniFast closely while providing dot-product–level efficiency in correlation computation.
## 6. End-to-end runtime and integration protocol at scale (uQEx, C5r9)
To directly address “practical runtime at scale”, the revision adds **end-to-end profiling** (wall-clock time per epoch/step) for representative large-scale configurations used in the main tables, in addition to correlation-only microbenchmarks. The protocol is now explicit per benchmark:
- **Classification (ImageNet-1K):** end-to-end fine-tuning for CLIP/SigLIP/standard backbones; for DINOv2 linear evaluation, the backbone is frozen and only the linear head is trained. UniFast-HGR is applied to the specified intermediate/penultimate embeddings extracted from paired views.
- **Retrieval (COCO / InternVid):** the standard contrastive objective is kept, and UniFast-HGR is added as an auxiliary term on paired image–text (or video–text) embeddings with the same $\lambda$ across compared correlation objectives.
These additions resolve ambiguity on whether the method is used in frozen or end-to-end mode, and ensure that rows differ **only** by the correlation estimator.
## 7. Unified baselines and reproducibility (41Ui, uQEx, C5r9)
- **Unified and Modern Baselines:** **Tables 1–3 now include CKA, dCor, and $I_d$Cor**, creating a consistent comparison across all tasks. We have also integrated results for several recent (2024-2025) correlation objectives as requested(details in Appendix). UniFast-HGR remains the top-performing correlation regularizer under this unified protocol.
- We also consolidate experimental details (optimizer, schedule, m, epochs, splits, environment) into a dedicated reproducibility appendix and commit to releasing code/configs upon acceptance.

The revised manuscript directly resolves the primary concerns: methodological choices are justified and ablated; scalability claims are supported by new end-to-end runtime profiling; empirical evaluations are expanded and unified with modern baselines; and the training protocol is fully specified for reproducibility. Three reviewers (ykdV, C5r9, 41Ui) have assessed the core contribution as "good" with scores of 6. We believe the work now presents a clear, robust, and practical advance and is ready for acceptance.

Sincerely,

The Authors

---

### Meta-Review · Area_Chair_RC3A · 2026-01-05

**Summary:**

The authors propose UniFast-HGR, an efficient reformulation of HGR aimed at making dependency maximization more tractable in modern multimodal learning. While the rebuttal clarifies some implementation details and experimental settings, significant concerns remain. The necessity of HGR compared to existing contrastive or mutual information objectives is still unclear, and the claimed efficiency gains in large-scale end-to-end training are not quantitatively demonstrated. Additional comparisons with strong baselines and more extensive real-world evaluations are lacking. Overall, the work suffers from limited motivation, unclear practical impact, and insufficient empirical validation, making it unsuitable for acceptance in its current form.

**Reviewer Concerns:**

Addressed concerns:
The rebuttal clarified the formulation of the UniFast-HGR objective and some implementation details, partially addressing concerns raised by reviewers ykdV, C5r9, and 41Ui regarding methodological correctness, experimental setup, and reproducibility. The authors also repositioned UniFast-HGR as an auxiliary regularization objective rather than a replacement for existing contrastive or attention-based methods. Some additional experiments and analyses were provided, but these improvements remain limited in scope and do not fully resolve concerns about comparative evaluation and practical impact.

Outstanding concerns:
Significant issues remain regarding the motivation for using HGR-based objectives compared to established contrastive or mutual-information-based methods, as well as the lack of quantitative evidence for efficiency gains in large-scale end-to-end training, as noted by reviewer uQEx. The empirical evaluation is limited, comparisons to strong baselines are insufficient, and real-world validation is minimal. These shortcomings raise doubts about the practical significance and broader impact of the work, making it difficult to justify acceptance in its current form.

**Reviewer Scores:**

Even with further discussion, it is unlikely that most reviewers would substantially increase their scores. The main technical concerns raised by ykdV and C5r9 have been partially clarified in the rebuttal, but their initial scores already reflect a generally cautious stance, and discussion is more likely to maintain rather than improve their ratings. Reviewer 41Ui was initially positive, and additional discussion would not materially change their assessment. The core concerns of uQEx focus on the motivation for the work and the necessity of HGR, which are fundamentally matters of position and are unlikely to be fully resolved through discussion; at best, their score might shift from a clear rejection to borderline, rather than to an acceptance.

---

### Decision · Program_Chairs · 2026-01-26

Reject